# The GRISLI-LSCE contribution to ISMIP6, Part 2: projections of the Antarctic ice sheet evolution by the end of the 21[st] century

Aurélien Quiquet[1] and Christophe Dumas[1]

[1]Laboratoire des Sciences du Climat et de l'Environnement (LSCE), UMR8212, CEA/CNRS-INSU/UVSQ, Gif-sur-Yvette Cedex, France

*Correspondence to:* A. Quiquet (aurelien.quiquet@lsce.ipsl.fr)

**Abstract.**

The Antarctic ice sheet's contribution to global sea level rise over the 21[st] century is of primary societal importance and remains largely uncertain as of yet. In particular, in the recent literature, the contribution of the Antarctic ice sheet by 2100 can be negative (sea level fall) by a few centimetres or positive (sea level rise) with some estimates above one metre. The Ice Sheet Model Intercomparison Project for CMIP6 aimed at reducing the uncertainties on the fate of the ice sheets in the future by gathering various ice-sheet models in a common framework. Here, we present the GRISLI-LSCE contribution to ISMIP6-Antarctica. We show that our model is strongly sensitive to the climate forcing used, with a contribution of the Antarctic ice sheet to global sea level rise by 2100 that ranges from -50 mm to +150 mm of sea level equivalent. Future oceanic warming leads to a decrease in thickness of the ice shelves resulting in grounding line retreat while increased surface mass balance partially mitigates or even overcompensates the dynamic ice sheet contribution to global sea level rise. Most of ice sheet changes over the next century are dampened under low greenhouse gas emission scenarios. Uncertainties related to sub-ice-shelf melt rates induce large differences in simulated grounding line retreat, confirming the importance of this process and its representation in ice-sheet models for projections of the Antarctic ice sheet's evolution.

## 1 Introduction

The Greenland and Antarctic ice sheets are now the largest source for the observed global mean sea level rise behind the thermosteric and the glacier contributions (Nerem et al., 2018). Given its size, the Antarctic ice sheet represents the largest single potential contributor in the future, as it represents 58 m of sea level rise if melted completey (Fretwell et al., 2013; Morlighem et al., 2020). While the ice sheet was probably in a quasi mass-equilibrium in the eighties (Rignot et al., 2019), it has since then lost ice at an accelerated pace, contributing to 7.6 mm to the global sea level rise over 1992-2017 (The IMBIE team, 2018). The largest changes are observed in West Antarctica with increased ice discharge (Gardner et al., 2018) and increased ice shelf mass loss (Paolo et al., 2015). These recent changes might have already triggered mechanical instabilities (Favier et al., 2014) that could led to an irreversible retreat of the grounding line over large sectors of the ice sheet. While the acceleration of mass loss is mostly associated with ocean warming, the increased precipitation related to climate change can partially mitigate the

ice sheet contribution to sea level rise in the future (Palerme et al., 2017; Medley and Thomas, 2019).

Despite significant advances in our understanding of ice-sheet dynamics (Pattyn et al., 2017), the projected sea level contribution of the Antarctic ice sheet in the future by numerical models remains largely uncertain (Oppenheimer et al., 2019). Overall, the uncertainties related to model formulation, parameter choice and external forcing, lead to a wide spread in the assessment of the magnitude of the Antarctic ice sheet contribution to global sea level rise by 2100, which can be either negative (sea level fall) by a few centimetres or positive (sea level rise) with some estimates above one metre (Golledge et al., 2015; Winkelmann et al., 2015; Ritz et al., 2015; DeConto and Pollard, 2016; Schlegel et al., 2018; Edwards et al., 2019; Bulthuis et al., 2019; Levermann et al., 2020; Seroussi et al., 2020). While the different ice-sheet models seem to respond consistently to atmospheric changes, oceanic changes translate instead into largely different model responses (Seroussi et al., 2019, 2020).

The Ice Sheet Model Intercomparison Project for CMIP6 (ISMIP6, Nowicki et al., 2016), endorsed by the Coupled Model Intercomparison Project – phase 6 (CMIP6), is an international effort that aims at providing estimates of the Greenland and Antarctic ice sheet contributions to global sea level rise by the end of the century. Such intercomparison of models is useful to reduce the uncertainties related to ice-sheet dynamics since a variety of ice-sheet models have participated in ISMIP6, spanning a range of model complexities and using various initialisation techniques to infer the initial conditions used for the projections. The analysis of the different responses amongst participating ice-sheet models is done in Goelzer et al. (2020) for the Greenland ice sheet and in Seroussi et al. (2020) for the Antarctic ice sheet. With the same ice-sheet model (GRISLI, Quiquet et al., 2018) and a similar ice-sheet initialisation procedure, we participated in both ISMIP6-Greenland and ISMIP6-Antarctica. This paper aims at presenting the GRISLI-LSCE contribution to ISMIP6-Antarctica in detail, while its companion paper (Quiquet and Dumas, 2020a) presents the ISMIP6-Greenland contribution. Thanks to a relatively low computational cost, we performed the full list of experiments of ISMIP6 described in (Nowicki et al., 2020), where Seroussi et al. (2020) only cover a subset of these experiments.

The analysis of a single model response to the different forcing scenarios presents some important added value with respect to the community paper of Seroussi et al. (2020). First, single model paper allows for a documentation of a specific model response to the forcings while this information can be buried in the community paper given the large material to cover. Second, the community paper is best suited for a quantification of the sensitivity of the projections to the choice of the ice-sheet model. In turn, the sensitivity to the climate forcing is better shown for individual ice-sheet model. Third, single model paper can provide a more complete information of model biases.

In Sec. 2 we describe the ice-sheet model used for the GRISLI-LSCE contribution and how the model has been initialised. In this section, we also present the ISMIP6 forcing methodology and we describe the complete list of experiments performed. The Antarctic ice sheet simulated by GRISLI for all the different experiments are presented in Sec. 3. Sec. 4 is a broader discussion

of these results and we conclude in Sec. 5

## 2 Methods

### 2.1 Model and initialisation

The experiments shown here were performed with the 3D thermo-mechanically coupled ice-sheet model GRISLI. Solving the mass and momentum conservation equations together with the heat equation, the model computes the evolution of the Antarctic ice sheet geometry and ice physical characteristics. The model is fully described in Quiquet et al. (2018) where the model has been shown to be capable of simulating grounding line migration of the Antarctic ice sheet at the glacial-interglacial timescale. For century timescales, with the same model version that the one used here, we participated to initMIP-Antarctica (Seroussi

et al., 2019), ABUMIP (Sun et al., 2020) and LarMIP (Levermann et al., 2020). Slightly earlier version of the model has been used to simulate the evolution of the Greenland ice sheet until 2100 and 2150 (Peano et al., 2017; Le clec'h et al., 2019a). In the following we only provide the main equations useful for the discussion of the model results.

In GRISLI, the ice sheet is only composed of incompressible ice with a constant and homogeneous density. The mass conser-

vation equation reads

$$\frac{\partial H}{\partial t} = M - \nabla \left( \overline{\boldsymbol{U}} H \right),$$  (1)

with $H$ the local ice thickness, $M$ the total mass balance and $\overline{\boldsymbol{U}}$ the vertically averaged horizontal velocity vector. $\nabla \left( \overline{\boldsymbol{U}} H \right)$ is thus the ice flux divergence.

Velocities are computed using asymptotic *shallow* zero-order approximations, namely the shallow ice approximation (SIA) and

the shallow shelf approximation (SSA). For the entire grid, the SSA is used as a sliding law (Bueler and Brown, 2009) and the total velocity results from the addition of the SIA and the SSA velocities as in Winkelmann et al. (2011). Floating ice shelves are assumed to have no friction at the base (SSA driven ice flux). Conversely, grounded cold based regions show an infinite friction (SIA driven ice flux). For temperate regions, we assume a linear basal friction (Weertman, 1957):

$$\boldsymbol{\tau_b} = -\beta \, \boldsymbol{U_b},$$  (2)

where $\tau_b$ is the basal drag, $\beta$ is the basal drag coefficient and $\boldsymbol{U_b}$ is the basal velocity. The basal drag coefficient is spatially variable but constant in time (except in specific cases such as during the inversion procedure).

As in most large-scale ice-sheet models, GRISLI uses a flow enhancement factor to artificially account for ice anisotropy (Quiquet et al., 2018). In the model, we specify the value of this enhancement factor for the SIA velocity and we use a fixed ratio to determine its smaller SSA counterpart. For the experiments presented here (except in Sec. 3.2.7), we use a flow enhancement

factor of 1 (no SIA enhancement) and a ratio close to 1 for the SSA (1.2:1).

Since the model is generally used at a coarse resolution (greater than 5 km), we use an analytical formulation of the flux at the

grounding line following either Schoof (2007) or Tsai et al. (2015). The sub-grid position of the grounding line is estimated with a linear extrapolation of the floatation criteria. From this sub-grid position of the grounding line, the ice flux from Schoof (2007) or Tsai et al. (2015) is extrapolated to the neighbouring velocity grid points. More details on this implementation is provided in Quiquet et al. (2018). Using a 40 km grid resolution the model was able to reproduce glacial-interglacial grounding
line migration in agreement with geological data (Quiquet et al., 2018). A 16 km version was also used to assess the importance of buttressing for grounding line stability in the ABUMIP intercomparison exercise Sun et al. (2020), where GRISLI shows an important grounding line retreat although amongst the lowest within the other participating models. Here, we use the analytical flux of Schoof (2007) at the grounding line.

Calving is based on a simple threshold criterion: the ice thickness at the front reaching a minimal value is automatically calved
if the upstream flux is not sufficient to maintain an ice thickness above this critical threshold. The minimal ice thickness is set to 200 m in the experiments presented here.

For model initialisation, we followed a similar approach as in the initMIP-Antarctica experiments (Seroussi et al., 2019). The initialisation procedure consists of an iterative method which aims at determining the geographical distribution of the basal drag
coefficient ($\beta$ in Eq. 2) that yields the minimal ice thickness error with respect to the observations. The procedure is described in Le clec'h et al. (2019b) and we only provide here a general description. We first compute an ice sheet thermal regime that is in equilibrium with the present-day climate forcing. To this aim, we run a 60 kyr experiment under a constant present-day climate forcing and impose a fixed topography. For this thermal equilibrium experiment, the basal drag coefficient comes from a previous model realisation (Levermann et al., 2020) and is left unchanged. Using the inferred thermal state at the end of the
60 kyr, we performed multiple 120-yr long experiments. Each iteration consists of a first step of 20 years with fixed grounding line position during which the basal drag coefficient is interactively adjusted on a yearly timestep so that it compensates the ice thickness error with respect to the observations (e.g. basal drag coefficient reduced for ice thickness overestimation). The second step is a 100 year long experiment with a freely evolving grounding line during which the basal drag coefficient remains at its last computed value during the first step. The ice thickness mismatch with respect to the observations at the end of the
100 simulated years is used to modify the basal drag coefficient for the next iteration. To do so, we modify the velocity so that the corrected vertically averaged velocity $\overline{U^{\mathrm{corr}}}$ is related to the simulated vertically averaged velocity $\overline{U}$ as

$$\overline{U^{\mathrm{corr}}} = \overline{U} \times \frac{H}{H^{\mathrm{obs}}}, \tag{3}$$

with $H$ and $H^{\mathrm{obs}}$ the simulated and the observed ice thickness. Only the basal velocity is corrected ($U_b^{\mathrm{corr}}$) when modifying the basal drag coefficient and we use the following relationship to infer the basal drag coefficient for the next step $\beta^{\mathrm{new}}$:

$$\beta^{\mathrm{new}} = \beta^{\mathrm{old}} \times \frac{U_b}{U_b^{\mathrm{corr}}}, \tag{4}$$

with $\beta^{\mathrm{old}}$ the basal drag coefficient of the previous step.

For the experiments shown here we have performed 15 iterations. At the end of the initialisation procedure, we use the last inferred basal drag coefficient together with the corresponding thermal state to run a relaxation experiment of 65 years with

a free evolving grounding line. The simulated ice sheet after this relaxation experiment is used as the initial condition for the historical experiment (*hist*, see Sec. 2.3). It should be noted that such initialisation procedure produces an ice sheet in quasi-equilibrium with the late-20th century mean climate state. By construction it does not simulate the accelerated mass loss observed in the last decades (Rignot et al., 2019).

Our reference ice thickness and bedrock topography is the Bedmap2 dataset (Fretwell et al., 2013). This dataset is used as the initial topography for the 65-yr relaxation experiment used to define the initial state for the historical simulation. The ice thickness in Bedmap2 is also used as a target for the iterative initialisation procedure. Our reference present-day surface mass balance comes from RACMO2.3p2 (van Wessem et al., 2018) averaged over 1979-2016. The reference present-day oceanic forcing used to compute the sub-ice-shelf melt rates (more details available in Sec. 2.2) is derived from a combination of observational datasets (Jourdain et al., 2020), averaged over 1995-2017. These reference atmospheric and oceanic forcings are used during the initialisation procedure and for the relaxation and control experiments (*ctrl* and *ctrl_proj*, see Sec. 2.3). The geothermal heat flux is taken from Shapiro and Ritzwoller (2004). The model is run on a Cartesian grid at 16 km resolution covering the Antarctic ice sheet using a polar stereographic projection. Glacial isostatic adjustment has been neglected in this work.

## 2.2 ISMIP6-Antarctica forcing methodology

The ISMIP6-Antarctica working group has elaborated and distributed atmospheric and oceanic forcings in addition to a detailed methodology on how to implement these forcings in individual ice-sheet models (Nowicki et al., 2020). Since we have strictly followed the suggested forcing methodology we only provide here the main principles and the reader is invited to refer to Nowicki et al. (2020) for more details.

For ice-sheet model projections, the ISMIP6-Antarctica working group has provided a set of yearly climate fields derived from various general circulation models (GCMs). The climate fields cover the 1950-2100 period.

- The atmospheric forcing consists of yearly surface mass balance and surface temperature (skin temperature) anomalies with respect to the 1995-2014 mean. The surface mass balance has been computed from the GCM outputs as the total precipitation minus the evaporation and runoff and regridded to the 16 km resolution grid. The anomalies have to be added on top of the reference present-day climatology.

- The oceanic forcing is the thermal forcing, i.e. the ambient temperature minus the ambient temperature at the freezing point. In the standard ISMIP6-Antarctica approach, which we follow with GRISLI, the thermal forcing is used to compute sub-ice-shelf melt rates using a non-local quadratic parametrisation as described in Jourdain et al. (2020). This parametrisation defines 16 sectors based on the Antarctic drainage basins extended into the open ocean. For each grid point $(x,y)$ of the model, belonging to a specific sector, the sub-ice-shelf melt rate $m$ is

$$m(x,y) = \gamma_0 \times K \times (TF(x,y,z_{\text{draft}}) + \delta T_{\text{sector}}) \times |\langle TF \rangle_{\text{draft} \in \text{sector}} + \delta T_{\text{sector}}| \qquad (5)$$

where $K$ is a constant that depends on physical properties of water, $TF(x, y, z_{\text{draft}})$ is the thermal forcing at the ice–ocean interface, $\langle TF \rangle_{\text{draft} \in \text{sector}}$ is the averaged thermal forcing for the ice shelves of the sector, and $\delta T_{\text{sector}}$ is a sector-specific temperature correction. $\gamma_0$ is a parameter calibrated to reproduce the observed melt rate in the observations for $\delta T_{\text{sector}} = 0°K$ . Once $\gamma_0$ is found, a $\delta T_{\text{sector}}$ correction is computed to reduce the sector-specific biases.

$\gamma_0$ is estimated in two different ways. In one approach, $\gamma_0$ is calibrated to reproduce the total Antarctic melt rate (Rignot et al., 2013; Depoorter et al., 2013). This version is labelled *MeanAnt* in Jourdain et al. (2020). An alternative calibration (labelled *PIGL*) consists in using a subset of the observational data, restricted to the Pine Island glacier sector. This is motivated by the fact that the Pine Island glacier has undergone a substantial grounding line retreat related to an increased sub-ice-shelf melting rates in the recent years (Jenkins et al., 2018). Also, there are dense observational data available in this sector. The *PIGL* calibration produces a higher melt rate response for a given change in thermal forcing than the *MeanAnt* calibration. Here, the experiments that used the *PIGL* calibration are labelled *PIGL* while all the other experiments use the *MeanAnt* calibration.

For both *MeanAnt* and *PIGL*, the $\gamma_0$ probabilistic distribution is computed with a random sampling of the melt rates in the observations. For each calibration, three possible values of $\gamma_0$ are thus given: the 5[th] percentile, the median and the 95[th] percentile. These results in different oceanic sensitivity to thermal forcing and are referred as *low*, *medium* and *high* oceanic sensitivity in this manuscript.

Surface melt can generate ice-shelf collapse through hydrofracturing (Scambos et al., 2009). These processes are poorly understood and generally not accounted for in large-scale ice-sheet models such as GRISLI. ISMIP6-Antarctica working groups have provided the participants with scenarios for ice-shelf collapse in the future following the methodology of Trusel et al. (2015). With these scenarios, the retreat in time of the ice-shelf front is imposed. These scenarios are not necessarily used and only the experiments labelled *shelf collapse* (hereafter *SC*) make use of them.

## 2.3 List of experiments

The ice-sheet state (i.e. ice thickness and internal thermomechanical conditions) at the end of the initialisation procedure (Sec. 2.1) is used as the initial condition for a control experiment *ctrl* and for the historical simulation *hist*. For the control experiment *ctrl*, the climate forcings (surface temperature, surface mass balance and thermal forcing) are left unchanged for the duration of the experiment at their present-day values used during the initialisation procedure (no anomaly is imposed). The *ctrl* experiment starts in January 1995 and ends in Decembre 2100, even though it uses a constant present-day climate forcing (RACMO2.3p2 averaged over 1979-2016). Instead, the historical simulation *hist* uses the time varying climate forcing described in Sec. 2.2 from January 1995 to Decembre 2014. Although it could have been possible to run multiple historical simulations for each GCM output available, it has been asked to participating models to run only one historical simulation using the NorESM1-M climate forcing. NorESM-1-M was chosen because it is one of the CMIP5 models that best reproduce

the present-day Antarctic climate change (Barthel et al., 2020).

The different ice-sheet projection experiments start in January 2015 and they are all branched from the end of the historical experiment *hist* (Decembre 2014). They end in Decembre 2100 (86 simulated years). The complete list of experiments in ISMIP6-Antarctica is shown in Tab. 1. Because few CMIP6 models were available when elaborating the ice-sheet forcing, most of the experiments make use of CMIP5 models. Four CMIP6 models are nonetheless used (Tier 2). Some climate models were run under two scenarios for future greenhouse gas evolution, a high emission scenario (RCP8.5 for CMIP5 models and SSP585 for CMIP6 models) and a low emission scenario (RCP2.6 for CMIP5 models and SSP126 for CMIP6 models). For each climate forcing, three experiments using different sub-ice-shelf melt rate sensitivity to temperature change (*low*, *medium* and *high*) are performed. In addition, the parametrisation of the sub-ice-shelf melt model calibrated against the Pine-Island glacier area (*PIGL*) is used for four CMIP5 models under RCP8.5. The ice-shelf collapse scenario related to hydrofracturing is also used for all the climate forcing under the high emission scenario. Finally, in order to disentangle the role of atmospheric versus oceanic forcing, a series of experiments also consists in using only one or the other of these forcings.

In order to allow for the interpretation of the model response to the forcings, a control experiment, *ctrl_proj*, has been performed in addition to the *ctrl* experiment. As in the *ctrl* experiment, the climate forcings remain constant with no anomaly with respect to the present-day climate used for the initialisation procedure. However, the *ctrl_proj* starts from the end of the historical simulation in January 2015 where the *ctrl* experiment uses the initial state instead. In doing so, the *ctrl_proj* experiment resembles a projection experiment, except that it uses no anomaly for the climate forcing.

## 3  Results

While the comparison of the various participating ice-sheet models response has been fully described in Seroussi et al. (2020), we aim here at describing the response of one individual model to the various forcings available in ISMIP6-Antarctica. A map of Antarctica with the names of the different regions discussed in the following is shown in Fig. 1.

### 3.1  Present-day simulated ice sheet

The map of ice thickness error with respect to the observations at the end of the historical simulation is shown in Fig. 2a. These errors are the results of ice thickness changes during the 65 years of relaxation at the end of the initialisation procedure and during the 20 years of the historical simulation. The differences appear relatively noisy since the model has a tendency to simulate smoother ice thickness gradients than observations. The differences over the East Antarctic plateau are smaller than a few metres but they increase towards the ice margins or in the vicinity of major ice streams (e.g. Amery ice-shelf tributaries). In East Antarctica, the Amery and Totten ice-shelf regions display the largest error where it can locally approach 500 metres. The ice thickness is generally overestimated in the the Amery region, while it is underestimated in the Totten region. While the errors are relatively localised in East Antarctica, they are more widespread in West Antarctica. There are large ice thickness

underestimations, locally reaching more than 200 metres, in the Getz ice-shelf region in the Amundsen sea and upstream the grounding line of the Filchner-Ronne ice shelf. The Pine Island glacier area shows an ice thickness overestimation of about 50 metres. Except for the Filchner ice shelf, the ice thickness of the ice shelf is slightly underestimated (error lower than 30 metres). The ice front of the Ross and Filchner-Ronne ice shelves is located about 80 km away from the observations. Overall, these discrepancies, integrated over the whole ice sheet, lead to an ice thickness root mean square error with respect to the observations of about 120 metres (5th lowest error amongst the 21 ISMIP6-Antarctica participating models).

The simulated surface velocity magnitude at the end of the historical simulation is shown in Fig. 3a. The model generally re-produces the pattern and the magnitude of the observed surface velocities, depicted in Fig. 3b, even if substantial errors remain (Fig. 3c). The largest errors are located in fast flowing areas and they can be positive (overestimation) or negative (underesti-mation). Surface velocities of the major tributaries of the Ross ice shelf (Mercer and Williams glaciers) and Filchner-Ronne ice shelf (Foundation glacier) are largely overestimated (locally up to a factor 4 with errors larger than 1000 m yr$^{-1}$). Conversely, there is a large underestimation of the ice velocity, locally greater than 1000 m yr$^{-1}$, for the Pine Island ice-shelf tributaries. The velocity errors for the grounded part of the ice sheet mostly explain the velocity errors for the floating ice shelves. Thus, the velocity in the Ross ice shelf is largely overestimated since its tributaries show generally a large ice velocity overestimation. The western part of the Ronne ice shelf shows an opposite behaviour with feeding glaciers showing a velocity underestimation. The Amery ice shelf is an exception: the grounded velocity errors are positive while their floating counterparts are negative. This ice shelf is narrow and very confined with a complex sub-ice-shelf melt rate pattern which makes it difficult to model for a large scale ice-sheet model at 16 km horizontal resolution. More generally, spatial resolution could explain most of velocity errors in the coastal regions where topography together with spatially variable surface mass balance and sub-ice-shelf melt exert a strong control on simulated velocities. Overall, the root mean square error with respect to the observations is about 270 m yr$^{-1}$ (3rd largest error amongst the 21 participating models). When computing the error for the logarithm of the velocity in order to reduce the importance of fast flowing regions with respect to slowly flowing regions, the performance of GRISLI with respect to the other participating models slightly improves (6th largest error). This suggests that the model shows the largest disagreement with respect to the observations in fast flowing regions. Our initialisation procedure aims at finding the basal drag coefficient that minimises the ice thickness error with respect to the observations but it does not have any constraints on the simulated velocities. As a result, it is not surprising that we obtain a low RMSE in ice thickness together with a larger RMSE in surface velocities with respect to other ice-sheet models that use the velocities in their initialisation procedure (e.g. JPL1_ISSM or UTAS_ElmerIce).

Even though our initialisation procedure aims at providing a simulated ice sheet in equilibrium with our reference present-day climate, a drift is nonetheless simulated at the century scale. Fig. 2b shows the ice thickness change from 2015 to 2100 in the control experiment *ctrl_proj*. The pattern of ice thickness change resembles the one of the ice thickness error with respect to observations (Fig. 2a). In particular the regions with the largest errors with respect to observations are the one producing the largest ice thickness change in the control simulation. The model drift over the 2015-2100 period can be explained for a

large part by the simulated velocity errors with respect to observations (Fig. 3c): thickening (e.g. Pine Island glacier region) is generally associated to an underestimation of the velocity while thinning (e.g. Filchner ice-shelf tributaries) is associated to an overestimation of the ice velocity. One exception is the Amery region in East Antarctica where the grounded velocities are overestimated while there is an increase in ice thickness in the control experiment. The ice thickening during the control exper-
5 iment could suggest an underestimation of the ice velocity, i.e. underestimation of the ice export, which seems in contradiction to the overestimation of the simulated ice velocity with respect to the observations. This inconsistency can be due to a surface mass balance overestimation in the forcing in this area. This overestimation could be corroborated by the fact that another regional climate model than the one used here simulates a surface mass balance 30% smaller than RACMO2.7 in the Amery region (Agosta et al., 2019). Because of compensating errors, the ice thickness change, integrated over the duration of the
10 control experiment, leads to a negligible total ice mass change (less than 1000 Gt). However, the ice volume above floatation shows a negative trend (Fig. 4) which means that there is a mass transfer from the grounded to the floating part of the ice sheet in the control experiment. The model drift in the control experiment *ctrl_proj* in terms of surface velocity is shown in Fig. 3. The velocity changes for the grounded areas are generally limited to a few metres per year except for some ice streams feeding the Ross and Filchner-Ronne ice shelves. Although more localised, the changes in Pine Island, Getz and Totten areas can be
larger than one hundred metres per year. Since the ice shelves show a larger velocity magnitude, they also show the largest absolute velocity changes (a few hundred metres locally).

## 3.2 Ice sheet evolution projections

### 3.2.1 Ice sheet evolution for CMIP5 models using RCP8.5

The evolution of the total ice mass change for the different CMIP5 models under the high emission scenario for greenhouse gases (RCP8.5) and using the sub-ice-shelf melt parametrisation calibrated over the Antarctic-wide dataset (*MeanAnt*) is shown in Fig. 4. The total ice mass (Fig. 4a) is decreasing for the six CMIP5 models and for most models there is an acceleration of ice mass loss in the course of the century. HadGEM2-ES produces the largest mass loss (about $300 \times 10^3$Gt in 2100) while CSIRO-Mk3 produces the smallest loss (lower then $50 \times 10^3$Gt). The sub-ice-shelf melt rate sensitivity to temperature change consti-
tutes an important source of uncertainty for the forcings that produce the largest mass loss: for NorESM1-M and HadGEM2-ES the differences between the *low* and *high* oceanic sensitivity corresponds to a mass difference of about $100 \times 10^3$Gt.

The volume change contributing to sea level rise (i.e. above floatation) shows a different evolution than the total ice mass (Fig. 4b). While the total mass change is always negative, the simulated Antarctic contribution to sea level rise in 2100 for
the CMIP5 models can be either positive, e.g. ~ 60 mm of sea level equivalent (mm SLE) for HadGEM2-ES, or negative, e.g. -45 mm SLE for CCSM4. This means that the ice shelf volume is shrinking for all forcings over the course of the century while the grounded ice volume can increase or reduce depending on the forcing used. In addition, except for the HadGEM2-ES forcing, the Antarctic contribution to global sea level rise is always smaller than for the control experiment under constant

present-day forcing. This suggests that the climate forcing computed from the GCMs in the future leads to a larger integrated total mass balance compared to our reference present-day mass balance. Another way to show this is to investigate the grounding line migration in the course of the century. In Fig. 5 we show the grounded ice extent evolution, which is an integrated indicator of grounding line migration. For all the projection experiments, the grounded ice extent is always smaller than in the

control experiment, and this extent decreases in the course of the century. Thus, even for models that produce an important grounded ice volume increase in the future (e.g. CCSM4), the grounded ice extent is decreasing. This can be only explained by an increase in surface mass balance over the grounded area. In fact, most GCMs simulate an increase in precipitation in Antarctica related to the projected warming. This increase in precipitation can be partly compensated by an increase in runoff and evaporation. However, overall, most GCMs produce an increase integrated surface mass balance in the future. The differ-

ence in terms of surface mass balance change amongst the GCMs explains the large spread in simulated Antarctic ice sheet contribution to global sea level rise. Fig. 6 shows the evolution of the surface mass balance (Fig. 6a) and basal mass balance (Fig. 6b) over the next century, integrated over the ice sheet, for the different climate forcings. Despite a considerable interannual variability, the surface mass balance is generally slightly increasing by 15% to 25% (400 to 900 Gt yr$^{-1}$ increase), except for HadGEM2-ES where it shows a slight decrease of about 200 Gt yr$^{-1}$. Instead, the basal melting underneath ice shelves is

increasing for the different GCMs leading to an increase in mass loss by about 100% (e.g. 1500 Gt yr$^{-1}$ increase for IPSL-CM5A-MR) to more than 200% (e.g. 5000 Gt yr$^{-1}$ increase for HadGEM2-ES). The lack of surface mass balance increase in HadGEM2-ES combined with an increase sub-ice-shelf melt rate explains why this forcing produces the largest Antarctic contribution to future sea level rise.

The spatial pattern of ice thickness change in 2100 with respect to 2015 for a selection of climate forcings is shown in Fig. 7. For this figure, in order to better illustrate the impact of the forcings, the projected ice thickness change has been corrected for the ice thickness change in the control experiment *ctrl_proj* (shown in Fig. 2b). Fig. 7a is for a forcing that produces a large increase in grounded ice volume (CESM2) under RCP8.5 while Fig. 7b is for a forcing that produces a reduction in both the total and the grounded ice volume (NorESM1-M). For both forcings, the Ross, Filchner-Ronne and Amery ice shelves

show ice thinning, amplified in NorESM1-M with respect to CESM2. However CESM2 shows a more pronounced thinning for the Larsen and Fimbul ice shelves, illustrating the spatial heterogeneity amongst the different forcings. Associated with the increased surface mass balance in the course of the century (Fig. 6a), CESM2 produces a widespread thickening of the grounded ice sheet. When using NorESM1-M this thickening is present to a lesser extent and compensated by the thinning that results from the grounding line retreat in some areas (Ross or Totten ice shelves for example). Our model does not simulate

substantial changes in the Pine Island glacier area. In this region, there is a thickening of the ice sheet during the control experiment (Fig. 2b) with underestimated surface velocities (Fig. 3c). These biases can be due to the inferred basal drag coefficient during the initialisation procedure that leads to an underestimation of the velocities. The linear friction law implemented in our model can also result in an underestimation of the velocity (Brondex et al., 2019). Finally, the biases can also be the result of the complex topographic setting that might not be well captured at 16 km. The underestimated ice sheet velocity at the

grounding line in this area, together with the thickening bias, result in a small sensitivity to oceanic warming. However, for

other intercomparison exercises we have shown that our model is able to produce a grounding line retreat in this area (Sun et al., 2020).

For the variety of climate forcing used, the Ross and Totten sectors are the ones that most frequently present grounding line retreat and inland thinning. The Filchner-Ronne sector presents also an ice shelf thickness decrease although associated with a limited grounding line retreat. This is consistent with the average response of the ISMIP6 participating models (Fig 6 in Seroussi et al., 2020). The lack of sensitivity of the Pine Island sector is also a feature common to other participating models since the standard deviation of ice thickness change in this area is very high (> 200 m).

### 3.2.2 Ice sheet evolution for CMIP6 models using SSP585

Because CMIP6 models have shown a larger climate sensitivity than their CMIP5 counterparts (Forster et al., 2020), it is interesting to compare the projected Antarctic ice sheet evolution under the CMIP6 forcings with respect to the CMIP5 experiments discussed previously. In Fig. 8, we show that the CMIP6 forcings produce an ice sheet evolution in the range of what we simulate with the CMIP5 forcings. Three models produce very little total ice mass change with an evolution very similar to the CCSM4 CMIP5 model. Only UkESM1 produces a relatively large total mass reduction (-230$\times 10^3$Gt) although not associated with a positive ice sheet contribution to sea level rise (about -10 mm SLE). Similarly to CMIP5 climate models, the CMIP6 models simulate an increase in the integrated surface mass balance (Fig. 6a) that partly compensate the mass loss due to sub-ice-shelf melting (Fig. 6b). Thus, the new generation of climate projections does not seem to support fundamentally different Antarctic evolution in the future with respect to the previous climate projections. However, only four CMIP6 models have been used in ISMIP6-Antarctica and this subset might be not representative for the whole ensemble.

### 3.2.3 Ice sheet evolution for RCP2.6 and SSP126

In Fig. 9 we show the total ice mass change under three climate models that have run for a high (RCP8.5 or SSP585) and a low (RCP2.6 or SSP126) emission scenario for greenhouse gases. The total mass loss is systematically smaller when using the low emission scenarios. The model that produces the largest mass loss, NorESM1-M, also shows the most pronounced response to the choice of the scenario. For this model, even if the volume loss contributing to global sea level rise remains almost unchanged, there is a drastic reduction in total ice mass loss when using the low emission scenario. In this case, the ice shelves are able to survive in the course of the century. For the other two models, IPSL-CM5A-MR and CNRM-CM6-1, the main consequence of the use of the low emission scenario instead of the high emission scenario is a reduction of the volume above floatation. This is related to the fact that most GCMs produce an increase surface mass balance for the high emission scenario induced by increased precipitation. Such effect is weaker in the low emission scenario. As a result, by the end of the century, the Antarctic ice sheet contribution to global sea level rise is larger (about 30 mm SLE) in the low emission scenario with respect to the high emission one. However, compared to the high emission scenario, the simulated total ice mass evolution using the low emission scenario is closer to the mass evolution of the control experiment. This means that, in this case, the

simulated ice sheet changes in the future are dampened with respect to an higher emission scenario.

The impact of the greenhouse gas scenario on the spatial distribution of ice thickness change across 2015-2100 is shown in Fig. 7. NorESM1-M using the RCP8.5 scenario (Fig. 7b) produces drastically thinner ice shelves than when using the RCP2.6 scenario (Fig. 7c). The Ross ice shelf is thus able to survive until the end of the century with minimal thickness change under the RCP2.6 scenario. The grounded parts of the ice sheet show an opposite response: the RCP2.6 scenario leads to almost no change in thickness whereas a slight widespread thickening is simulated under RCP8.5, related to increased surface mass balance.

In Seroussi et al. (2020), two climate forcings (NorESM1-M and IPSL-CM5A-MR) were evaluated for both the RCP2.6 and the RCP8.5. The simulated contribution to sea level rise in the ISMIP6 ensemble is very similar to the GRISLI response: no change in grounded ice mass for NorESM1-M but an increase in grounded ice mass for IPSL-CM5A-MR under RCP8.5 with respect to RCP2.6. CNRM-CM6-1 shows a response similar to the one of the IPSL-CM5A-MR since the grounded ice mass is increasing under the SSP585 with respect to the SSP126.

### 3.2.4   Ice sheet evolution using the Pine-Island glacier calibrated sub-ice-shelf melt parametrisation

The computation of the sub-ice-shelf melt rate in ice-sheet models is one of the largest source of uncertainty. The standard approach in ISMIP6-Antarctica is a parametrisation tuned to reproduce a combination of observational datasets (Jourdain et al., 2020). However, the choice of the dataset used to calibrate the parametrisation can lead to substantial differences in the sub-ice-shelf melt model. Fig. 10 shows the simulated total ice mass change when using the sub-ice-shelf melt parametrisation calibrated to reproduce the mean Antarctic melt rate (reference, *MeanAnt*) or calibrated to reproduce the Pine Island's grounding line melt rate (*PIGL*). The *PIGL* calibration produces higher melt rates and much greater mass loss than the reference calibration. For the medium oceanic sensitivity, the use of the *PIGL* calibration leads to an additional total mass loss of 200 to $300\times10^3$Gt and an additional contribution to global sea level rise of about 40 to 50 mm SLE with respect to the *MeanAnt* calibration. In addition, with the *PIGL* calibration, the model shows a much larger sensitivity to the oceanic forcing as the difference from a low to a high oceanic sensitivity can be as large as $350\times10^3$Gt (100 mm SLE) when using the CCSM4 forcing.

Amongst the different experiments, the NorESM1-M under RCP8.5 using the *PIGL* calibration for the sub-ice-shelf melt rate with a high oceanic sensitivity produces the largest Antarctic contribution to global sea level rise by 2100. The spatial distribution of ice thickness change over 2015-2100 for this experiment is shown in Fig. 7d. The pattern is similar to the one obtained with the reference sub-ice-shelf melt model (Fig. 7b) but with a much larger decrease in ice thickness. In particular, the grounded line retreats much further inland in the Ross and Filchner-Ronne sectors when using the *PIGL* calibrated sub-ice-shelf melt model with the high oceanic sensitivity. This larger grounding line retreat is also visible in Fig. 5 which shows the

grounded ice extent evolution for *MeanAnt* (plain lines) and *PIGL* (dashed lines) for three climate forcings.

### 3.2.5  Ice sheet evolution using the ice-shelf collapse scenario

Fig. 11 shows the impact of the imposed ice-shelf collapse scenario on the total ice mass evolution when using different GCM forcings. Such scenarios lead to an increase in the total mass loss (Fig. 11a) but have, most of the time, a small impact on the ice volume contributing to global sea level rise (less than 16 mm SLE in 2100, Fig. 11b). This means that the ice-shelf collapse scenarios mostly impact the floating ice volume but, on the century time scale, they do not imply a destabilisation of the grounded ice sheet in our model. The largest response is obtained for CNRM-ESM2 and CNRM-CM6-1. These models show a limited sub-ice-shelf melt (Fig. 6b) and one of the smallest ice mass loss in the future (Fig. 8a). Thus, they produce a large ice-shelf extent with respect to the other climate models. CNRM-ESM2 and CNRM-CM6-1 also simulate a pronounced atmospheric warming in the future (Nowicki et al., 2020). The atmospheric warming together with the large ice-shelf extent explain why the CNRM-ESM2 and CNRM-CM6-1 models show the largest mass loss resulting from ice-shelf collapse. Overall, the ice shelf collapse scenario systematically induces a decrease in the ice shelf extent. For example, when using CCSM4 under the RCP8.5 the ice shelf extent decreases by 86 000 $km^2$ from 2015 to 2100, but it decreases by 240 000 $km^2$ with the ice shelf collapse scenario (extent loss 2.8 times larger).

The impact of the ice shelf collapse scenario on the sea level contribution ranges from -8 to +17 mm SLE. This range is much smaller than the range of the simulated sea level contribution for the different climate models (-50 to 70 mm SLE). Surprisingly, for some models, the ice shelf collapse scenario contributes negatively to the sea level contribution (e.g. UKESM1-0-LL). This is most probably due to local non-linearities of grounding line dynamics. However this effect is limited to small changes in the grounded volume.

A greater sensitivity to this process has been reported in Seroussi et al. (2020), although associated with a wide spread of responses amongst participating models. In terms of ice shelf extent loss, Seroussi et al. (2020) reported a loss 6 times larger with the ice shelf collapse scenario (66 000 $km^2$ compared to 11 000 $km^2$) for CCSM4 under RCP8.5. However, the numbers in Seroussi et al. (2020) are much smaller than the one in GRISLI (240 000 and 86 000 $km^2$ with and without the shelf collapse scenario, respectively) suggesting a high sensitivity of the ice shelf extent in GRISLI to the oceanic perturbation. This might explain why the ice shelf collapse has a relatively lower impact on the ice shelf extent. However, Seroussi et al. (2020) also reported a larger impact of the ice shelf collapse scenario on the volume change contributing to sea level rise (multi-model average of 28 mm SLE in 2100 under the CCSM4 forcing). This can indicate a low sensitivity of the grounding line retreat in GRISLI compared to the other participating models. However, it can also be linked to the local model biases. In fact, for most climate models, the retreat masks by 2100 have removed the ice shelves in the Peninsula and in the Pine Island sectors, but affect only very marginally the other ice shelves. In the standard experiments, these sectors show a low sensitivity to the oceanic forcing. In fact, even under the strongest oceanic forcings, GRISLI shows there a limited grounding line retreat. This

suggests that the buttressing force is not the reason why the model does not retreat in these sectors. Instead, it is most likely the topographic biases in the initial state that made the model weakly sensitive to the oceanic conditions. Using a different initial state, we have shown in a recent intercomparison exercise (ABUMIP, Sun et al., 2020) that we were able to simulate large grounding line retreats when the buttressing induced by the ice shelves is removed, although amongst the lowest within the other participating models (3[rd] lowest ice volume change with respect to the control experiment in 500 years, out of 15 participating models).

### 3.2.6 Role of atmospheric versus oceanic forcing

Future global warming has ambivalent impacts on the evolution of the Antarctic ice sheet. On the one hand, the Southern Ocean is expected to warm in the future, leading to ice shelf thinning and calving eventually associated to grounding line destabilisation. On the other hand, the increase in moisture content associated with atmospheric warming can lead to increased surface mass balance and thickening of the ice sheet. To disentangle the respective role of the oceanic forcing with respect to the atmospheric forcing, we have run the ice-sheet model for four climate forcings using alternatively only one or the other of the forcing (ocean only, OO, or atmosphere only, AO). The results in terms of total mass change is shown in Fig. 12. The AO experiments produce an increase in total ice mass where the OO experiments show a decrease (Fig. 12a). The Antarctic contribution to global sea level rise is smaller than the control experiment *ctrl_proj* for the AO experiments while the OO experiments produce a contribution relatively close to the control experiment *ctrl_proj*, although slightly larger. The CCSM4 model produces the largest surface mass balance increase (Fig. 6a). Interestingly, the Antarctic contribution to sea level rise with this model is almost identical when using the full forcing (Fig. 4b) or when using the atmospheric forcing only (Fig. 12b). This suggests a negligible role of the ocean for this model to explain the Antarctic ice sheet contribution to sea level rise in the future. To a lesser extent this is also the case for the MIROC-ESM-CHEM model. Conversely, the total ice mass change (Fig. 12a) mostly reflects the mass loss from the ice shelves which respond primarily to the oceanic forcing. The ice-shelf mass loss in the OO experiments can be large with an important acceleration in the last 20 years of the century. This late response might be a reason why the volume above floatation is not drastically different from the control experiment in the OO experiments.

### 3.2.7 Simulated change in ice dynamics

The ice-sheet surface velocity change in 2100 with respect to 2015 using the NorESM1-M climate forcing under RCP8.5 with the medium oceanic sensitivity is shown in Fig. 13a. Associated with ice thinning (Fig. 7b), the remaining ice shelves show a large decrease in surface velocity. Modelled grounded ice surface velocity changes are limited with the notable exception of the ice streams feeding the Ross ice shelf that show a substantial acceleration (several hundred metres per year). The acceleration in this area is due to the grounding line retreat simulated by the model under this climate scenario. The pattern of simulated ice velocity change is consistent with results from other ice-sheet models (Seroussi et al., 2020) and remains similar for the other forcings: decreased ice-shelf velocity and increased grounded velocity only for scenarios that produce a grounding line retreat

in the future.

Another way to quantify the dynamic changes over this century is to integrate in time the mass conservation equation (Eq. 1). In doing so, the total ice thickness change from 2015 to 2100 is the superposition of two terms of different causes: the integral of the mass balance related to climate forcings (calving and surface and basal mass balance) and the integral of the ice flux divergence. The integral of the ice flux divergence can be seen as the dynamical contribution to ice thickness change. Such dynamical contribution is shown in Fig. 13b for the NorESM1-M climate forcing with the medium oceanic sensitivity. Generally the dynamical contribution follows the simulated change in surface velocity. In West Antarctica, the dynamical contribution has a strong spatial variability. It can reach up to more than 50 metres decrease in ice thickness and as such explains most of the simulated ice thickness change shown in Fig. 7b. In East Antarctica there is a widespread very small (a few centimetres) negative dynamical contribution to ice thickness change (ice thinning) that somehow moderates the ice thickening due to increased surface mass balance.

To further assess the sensitivity of the simulated ice sheet evolution to the mechanical parameters used in the model, we performed a set of additional sensitivity experiments. In these new experiments, we apply a uniform perturbation of either the basal drag coefficient (Eq. 2) or the SIA flow enhancement factor. These perturbations are imposed abruptly at the end of the year 2045, in order to mimic a potential change of these parameters in the course of the century. The timing of these perturbations is somewhat arbitrary: not too close from the start of the projections but also not too late so that they affect the ice sheet evolution to 2100. We perform perturbed control experiments *ctrl_proj* and perturbed projections using the NorESM1-M climate forcing under RCP8.5 with a medium oceanic sensitivity. Fig. 14 shows the mass change in 2100 for the perturbed experiments with respect to their unperturbed counterpart (shown in Sec. 3.2.1). Fig. 14a,b is for a basal drag coefficient perturbation that starts from +100% (i.e. a doubling of the base value) to -90% (i.e. a reduction to 10% of the base value). Fig. 14c,d shows the effect of changing the value of the enhancement factor from 0.4 to 6 (1 being the standard value). The perturbed control experiments are used here to define a range of acceptable perturbations. Thus, in Fig. 14, the vertical grey band shows the range of perturbations that implies a 0.15% total mass change in the perturbed *ctrl_proj* with respect to the standard *ctrl_proj*. 0.15% has been chosen as it represents one tenth of the mass loss simulated using NorESM1-M under RCP8.5 with a medium oceanic sensitivity. For the basal drag coefficient, the acceptable perturbations lead to an additional sea level contribution ranging from about -30 to +30 mm SLE, with respect to the unperturbed NorESM1-M under RCP8.5 experiment that produces a 20 mm SLE in 2100. The perturbation produces thus considerable ice sheet changes. For the enhancement factor, the effect of the perturbation is even larger as it ranges from -50 to +50 mm SLE. These sensitivity experiments show that any change in the Antarctic ice-sheet mechanical properties (basal dragging or ice flow) in the course of the century can have a substantial impact on the ice sheet contribution to sea level rise. The total mass change is relatively less impacted by the perturbations. The perturbations induce a change in total mass of $-12 \times 10^3$ to $+12 \times 10^3$ Gt for the basal drag coefficient and of $-30 \times 10^3$ to $+25 \times 10^3$ Gt for the enhancement factor, with respect to the mass loss in 2100 of $-165 \times 10^3$ Gt obtained with the unperturbed NorESM1-M under RCP8.5 experiment. The total ice mass is less impacted by the perturbations than the mass contributing to sea level rise because the ice

shelves respond first to the increase sub-ice-shelf melt rate.

These simple sensitivity experiments can also be used to quantify the importance of the choice of the mechanical parameters for the projections. For the basal drag coefficient, the perturbations lead to a change in the sea level contribution that is almost identical for the projection experiments and for the control experiment. This means that the effect of climate change is not amplified for different values of the basal drag coefficient. As a result, with our model, the projected contribution to sea level rise is only weakly affected by the choice of the basal drag coefficient. For the enhancement factor, this does not hold: a larger (respectively smaller) enhancement factor leads to larger (respectively smaller) ice sheet contribution to sea level rise. However, if the difference can be as large as 50 mm SLE for an enhancement factor of 4, it is nonetheless small in the vicinity of the reference value of 1.

## 4   Discussion

Amongst the different experiments, the largest contribution by 2100 is 150 mm SLE (NorESM1-M *PIGL* with a high oceanic sensitivity) while most experiments produce a contribution no greater than 80 mm SLE. Thus, it appears that the contribution of the Antarctic ice sheet to global sea level rise simulated by GRISLI is relatively limited. Since ISMIP6-Antarctica was a large intercomparison exercise that involved 13 research groups and 21 model versions, it is useful to compare these numbers with the ISMIP6-Antarctica ensemble. For this ensemble, using a medium oceanic sensitivity, HadGEM2-ES produces the largest mass loss with an ensemble mean of 96 mm SLE and CCSM4 produces the largest mass gain with an ensemble mean of -37 mm SLE. Although GRISLI does not stand up as an outlier within the ISMIP6 ensemble, it shows a more limited sea level contribution with 58 mm SLE for HadGEM2-ES and -45 mm SLE for CCSM4. This could suggest a moderate sensitivity of the grounding line migration in response to the oceanic forcing when compared to the other ice sheet models. However, it is important to note that some outliers are largely influencing the ISMIP6-Antarctica ensemble mean towards higher contributions. In particular, some ice sheet models that do not use the standard ISMIP6 approach to compute sub-shelf melting (*open* experiments) produce much higher ice sheet mass loss. Notably, for NorESM1-M (RCP8.5 medium oceanic sensitivity), ULB_FETISH32_open, ULB_FETISH16_open, VUB_PISM_open and NCAR_CISM_open simulate a 2100 mass loss ranging from 72 mm SLE to 166 mm SLE where all the other models show an ensemble mean close to 0 mm SLE. In addition, when models use both the *standard* and the *open* approach to compute the sub-shelf melting, the *open* approach tends to produce much higher mass loss (NCAR_CISM, UCIJPL_ISSM, ULB_FETISH32, ULB_FETISH16). Thus, it seems that the consideration of how the different groups have implemented this process is crucial to understand the multi-model spread. When we consider only the models that use the *standard* approach, GRISLI shows a mass loss much closer to the ensemble mean. However, it is not excluded that GRISLI shows a relatively low oceanic sensitivity. It is for example unable to simulate any substantial grounding line retreat in the Pine Island glacier area for the different climate scenarios tested here, even though this could be linked to initialisation biases that induce an ice thickening in this area in the control experiment. Also, in the

ABUMIP intercomparison exercise (Sun et al., 2020), GRISLI shows one of the lowest grounding line retreat due to the loss of buttressing (3$^{rd}$ lowest ice loss in 500 years with respect to the control, out of 15 participating models). Sun et al. (2020) suggested that plastic friction laws produce greater grounding line sensitivity than linear friction law as the one used here. This was also suggested by Brondex et al. (2019). A foreseen improvement of our ice sheet model will be the implementation of various friction laws to better assess the sensitivity of grounding line dynamics to this process.

Beyond GRISLI, the ISMIP6-Antarctica ensemble mean is low (e.g. below 30 mm SLE for NorESM1-M under RCP8.5). A relatively moderate Antarctic ice sheet contribution to future sea level rise by 2100 has also been suggested in other studies since the IPCC special report on the ocean and cryosphere in a changing climate (Oppenheimer et al., 2019) reported a range from 30 to 280 mm SLE (RCP8.5). However, this seems nonetheless in contradiction with the acceleration in mass loss reported by modern observational techniques Rignot et al. (2019). One reason for this disagreement is that most models participating in ISMIP6, including GRISLI, use some kind of data assimilation procedure that produces an ice sheet initial condition in quasi-equilibrium with present-day forcing. This methodology is thus not suited to reproduce the recent acceleration in mass loss, particularly large in West Antarctica where it has been estimated to 48 Gt yr$^{-1}$ per decade for 1979-2017 (Rignot et al., 2019). For example, a simple cumulative value of the observed 2012-2017 loss rate (219 Gt yr$^{-1}$, The IMBIE team, 2018) from 2015 to 2100 will result in an Antarctic ice sheet contribution to sea level rise of 52 cm SLE. This number is much greater than the simulated contributions by GRISLI and more generally, it is much greater than any ISMIP6-Antarctica participating model simulated contribution. This highlights the importance of initial conditions for century scale projections. Assimilation of surface velocities in transient ice-sheet simulations are promising methodologies to overcome the limitations inherent to methods that assume steady state (Gillet-Chaulet, 2020). However, they require a complex modelling framework not currently implemented in our ice-sheet model. In future developments of our model, we plan to modify the target of the inversion procedure by adding the recent observed ice thickness changes to the observed ice thickness. This would provide a more realistic initial state for the projections.

The GRISLI ice-sheet model, similarly to other ISMIP6-Antarctica participating models, simulate an ice sheet contribution to global sea level in 2100 that can be either positive or negative, depending on the climate forcing used. This is related to the fact that the climate models simulate an increase in surface mass balance in the future over Antarctica. An important difference with ISMIP6-Greenland forcing methodology lies in the fact that the atmospheric forcing is much more simplified in ISMIP6-Antarctica. The ISMIP6-Greenland atmospheric scenarios has been elaborated from a regional climate model forced at its boundary by the different GCMs. The atmospheric forcing fields (namely surface temperature and surface mass balance anomalies) are further corrected by the surface elevation changes using time-evolving vertical gradients computed from the regional climate model. Such approach is much more computationally expensive since it requires multiple regional climate model simulations. For example, the MAR regional climate model (Agosta et al., 2019) requires about 15 days to compute 100 years (C. Agosta, personal communication). That is why this approach has been discarded so far for the Antarctic ice sheet where the GCMs anomalies are used directly with no downscaling with a regional climate model and no vertical correction.

The use of an approach similar to ISMIP6-Greenland would be a significant step forward for the next exercise for Antarctica given the importance of the atmospheric forcing for the Antarctic contribution to future sea level rise.

While the atmospheric forcing is an important driver for the Antarctic evolution, the oceanic forcing remains the major source of uncertainty for future projections. Thus, using a different calibration strategy, the *PIGL* sub-ice-shelf melt model produce a much larger ice sheet retreat than the standard calibration *MeanAnt*. In addition, Seroussi et al. (2020) also show that the ice-sheet models that use their own approach to compute the sub-ice-shelf melt in place of the standard ISMIP6-Antarctica melt model are the models that produce generally the largest Antarctic contribution to future sea level rise. Thus, the participating models that use the standard approach all simulate a loss in ice volume above floatation lower than 40 mm SLE in 2100 using NorESM1-M under RCP8.5 with a medium oceanic sensitivity. At the same time, four models that use their own approach simulate a much greater loss, ranging from about 72 to 166 mm SLE, when using forcings elaborated from the same climate model realisations. This highlights the need for a better understanding of this process, since the various parametrisations used in ice-sheet models lead to largely different simulated sub-ice-shelf melt rates (Favier et al., 2019).

## 5 Conclusions

In this paper, we have presented the GRISLI-LSCE contribution to ISMIP6-Antarctica, providing the means to investigate the impact of the climate forcing on one individual ice-sheet model. We showed that the total mass change simulated by 2100 is strongly dependant on the general circulation model used to force the ice-sheet model. On the one hand, the total ice mass is decreasing in the course of the century for all the climate forcings evaluated, primarily because of ice-shelf mass loss. The mass loss can be as low as $100 \times 10^3$Gt to as high as $700 \times 10^3$Gt. On the other hand, the ice volume contributing to sea level rise can be either positive (sea level rise) or negative (see level fall). We simulate a range of ice-sheet contributions to global sea level rise by 2100 from about -50 mm SLE to +150 mm SLE. Increased surface mass balance simulated by most climate models in the future tend to increase the grounded ice volume, partly mitigating or over-compensating the effect of loss of buttressing due to ice-shelf melt. By the end of the century, we simulate the largest changes in ice thickness and ice dynamics in the Filchner-Ronne and Ross basins with only moderate changes elsewhere. The geographical pattern of these changes remains mostly consistent amongst the different climate forcings. The CMIP6 climate models used for ISMIP6 do not drastically change the simulated ice-sheet volume in the future with respect to the CMIP5 models. Under low greenhouse gas emission scenarios, the Antarctic ice sheet exhibits much less ice mass changes suggesting that the ice-sheet mass loss could be mitigated with a reduction in greenhouse gas emission. The oceanic forcing is a major source of uncertainty since the use of the melt model calibrated against the Pine-Island glacier data instead of the standard calibration produces a much faster ice-shelf retreat and, as a result, a larger ice sheet contribution to sea level rise in the future. This process has to be carefully assessed when performing future projections of the Antarctic ice sheet. Finally, with additional simple sensitivity tests we have shown that the simulated

ice sheet contribution to sea level rise by 2100 could be largely affected by changes in ice-sheet mechanical properties such as basal dragging. Given the weak understanding on such processes, they could also represent a large source of uncertainty.

## 6  Data availability

The GRISLI outputs from the experiments described in this paper are available on the Zenodo repository with digital object identifier 10.5281/zenodo.3819782 (Quiquet and Dumas, 2020b). The outputs in the Zenodo repository are the standard GRISLI outputs on the native 16 km grid and, as a result, they may slightly differ from the post-processed outputs available on the official CMIP6 archive on the Earth System Grid Federation (ESGF). In order to document CMIP6's scientific impact and enable ongoing support of CMIP, users are obligated to acknowledge CMIP6, the participating modelling groups, and the ESGF centres (see details on the CMIP Panel website at http://www.wcrp-climate.org/index.php/wgcm-cmip/about-cmip). The forcing datasets are available through the ISMIP6 wiki (http://www.climate-cryosphere.org/wiki/index.php?title=ISMIP6_wiki_page, last access: 1 February 2021).

*Acknowledgements.* We thank the Climate and Cryosphere (CliC) effort, which provided support for ISMIP6 through sponsoring of workshops, hosting the ISMIP6 website and wiki, and promoted ISMIP6. We acknowledge the World Climate Research Programme, which, through its Working Group on Coupled Modelling, coordinated and promoted CMIP5 and CMIP6. We thank the climate modeling groups for producing and making available their model output, the Earth System Grid Federation (ESGF) for archiving the CMIP data and providing access, the University at Buffalo for ISMIP6 data distribution and upload, and the multiple funding agencies who support CMIP5 and CMIP6 and ESGF. We thank the ISMIP6 steering committee, the ISMIP6 model selection group and ISMIP6 dataset preparation group for their continuous engagement in defining ISMIP6. This is ISMIP6 contribution No 24.

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

**Table 1.** List of ISMIP6-Antarctica experiments performed in this work. The three oceanic sensitivities are low, medium (med) and high. The experiments that use the sub-shelf melt parametrisation calibrated against the Pine Island glacier data are labelled *PIGL*. The experiments that use the imposed ice-shelf collapse scenario due to hydrofracturing are labelled *SC*.

| exp_id | scenario | GCM | Ocean | |
|--------|----------|-----|-------|---|
| exp05 | RCP8.5 | NorESM1-M | Med | |
| exp06 | RCP8.5 | MIROC-ESM-CHEM | Med | |
| exp07 | RCP2.6 | NorESM1-M | Med | |
| exp08 | RCP8.5 | CCSM4 | Med | Core experiments – Tier 1 |
| exp09 | RCP8.5 | NorESM1-M | High | |
| exp10 | RCP8.5 | NorESM1-M | Low | |
| exp12 | RCP8.5 | CCSM4 | *SC* Med | |
| exp13 | RCP8.5 | NorESM1-M | *PIGL* Med | |
| expa05 | RCP8.5 | HadGEM2-RS | Med | |
| expa06 | RCP8.5 | CSIRO-Mk3 | Med | Extended ensemble – Tier 2 |
| expa07 | RCP8.5 | IPSL-CM5-MR | Med | |
| expa08 | RCP2.6 | IPSL-CM5-MR | Med | |
| expb06 | SSP585 | CNRM-CM6-1 | Med | |
| expb07 | SSP126 | CNRM-CM6-1 | Med | CMIP6 extension – Tier 2 |
| expb08 | SSP585 | UKESM1-0-LL | Med | |
| expb09 | SSP585 | CESM2 | Med | |
| expb10 | SSP585 | CNRM-ESM2-1 | Med | |
| expc01 | RCP8.5 | NorESM1-M AO | Med | |
| expc03 | RCP8.5 | NorESM1-M OO | Med | Ocean only (OO) and Atmos. only (AO) – Tier 3 |
| expc04 | RCP8.5 | MIROC-ESM-CHEM AO | Med | |
| expc06 | RCP8.5 | MIROC-ESM-CHEM OO | Med | |
| expc07 | RCP2.6 | NorESM1-M AO | Med | |
| expc09 | RCP2.6 | NorESM1-M OO | Med | |
| expc10 | RCP8.5 | CCSM4 AO | Med | |
| expc12 | RCP8.5 | CCSM4 OO | Med | |

| exp_id | scenario | GCM | Ocean | |
|--------|----------|-----|-------|---|
| expd01 | RCP8.5 | MIROC-ESM-CHEM | High | |
| expd02 | RCP8.5 | MIROC-ESM-CHEM | Low | |
| expd03 | RCP2.6 | NorESM1-M | High | |
| expd04 | RCP2.6 | NorESM1-M | Low | |
| expd05 | RCP8.5 | CCSM4 | High | |
| expd06 | RCP8.5 | CCSM4 | Low | |
| expd07 | RCP8.5 | HadGEM2-RS | High | |
| expd08 | RCP8.5 | HadGEM2-RS | Low | |
| expd09 | RCP8.5 | CSIRO-Mk3 | High | |
| expd10 | RCP8.5 | CSIRO-Mk3 | Low | |
| expd11 | RCP8.5 | IPSL-CM5-MR | High | |
| expd12 | RCP8.5 | IPSL-CM5-MR | Low | |
| expd13 | SSP585 | CNRM-CM6-1 | High | Ocean sensitivity – Tier 3 |
| expd14 | SSP585 | CNRM-CM6-1 | Low | |
| expd15 | SSP585 | UKESM1-0-LL | High | |
| expd16 | SSP585 | UKESM1-0-LL | Low | |
| expd17 | SSP585 | CESM2 | High | |
| expd18 | SSP585 | CESM2 | Low | |
| expd51 | RCP8.5 | NorESM1-M | *PIGL* Low | |
| expd52 | RCP8.5 | NorESM1-M | *PIGL* High | |
| expd53 | RCP8.5 | MIROC-ESM-CHEM | *PIGL* Med | |
| expd54 | RCP8.5 | MIROC-ESM-CHEM | *PIGL* Low | |
| expd55 | RCP8.5 | MIROC-ESM-CHEM | *PIGL* High | |
| expd56 | RCP8.5 | CCSM4 | *PIGL* Med | |
| expd57 | RCP8.5 | CCSM4 | *PIGL* Low | |
| expd58 | RCP8.5 | CCSM4 | *PIGL* High | |
| expe06 | RCP8.5 | NorESM1-M | *SC* Med | |
| expe07 | RCP8.5 | MIROC-ESM-CHEM | *SC* Med | |
| expe08 | RCP8.5 | HadGEM2-RS | *SC* Med | |
| expe09 | RCP8.5 | CSIRO-Mk3 | *SC* Med | Ice shelf collapse – Tier 3 |
| expe10 | RCP8.5 | IPSL-CM5-MR | *SC* Med | |
| expe15 | SSP585 | CNRM-CM6-1 | *SC* Med | |
| expe16 | SSP585 | UKESM1-0-LL | *SC* Med | |
| expe17 | SSP585 | CESM2 | *SC* Med | |
| expe18 | SSP585 | CNRM-ESM2-1 | *SC* Med | |

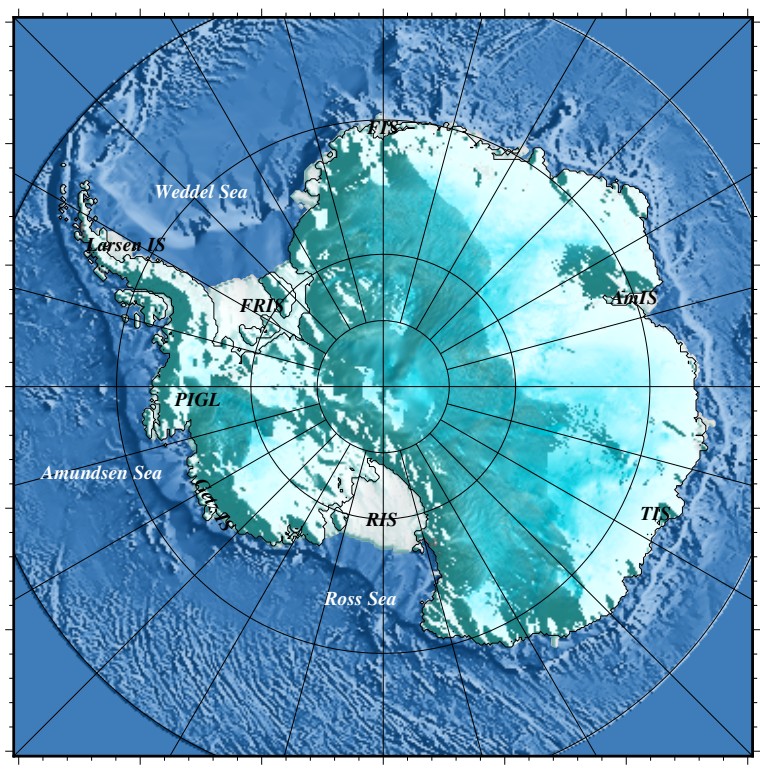

**Figure 1.** The Antarctic ice sheet with the major ice shelves discussed in the text: Larsen ice shelf, Filchner-Ronne ice shelf (FRIS), Pine Island glacier ice shelf (PIGL), Getz ice shelf, Ross ice shelf (RIS), Totten ice shelf (TIS), Amery ice shelf (AmIS) and Fimbul ice shelf (FIS).

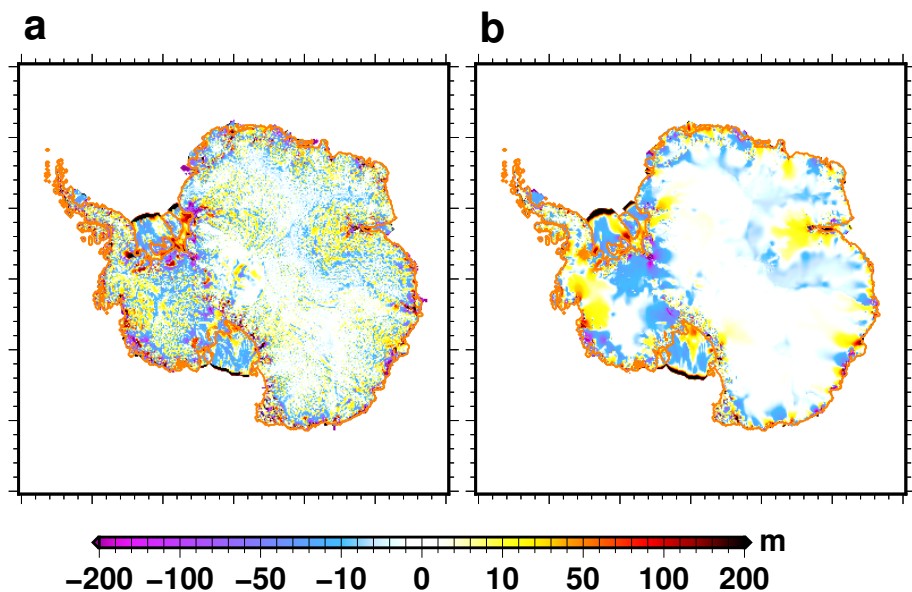

**Figure 2.** Ice thickness difference: **(a)** end of the historical experiment *hist* with respect to observations (Fretwell et al., 2013); **(b)** end of the control experiment *ctrl_proj* with respect to the end of the historical experiment *hist*. The orange line shows the present-day grounded line. The Pearson correlation coefficient between (a) and (b) is 0.24.

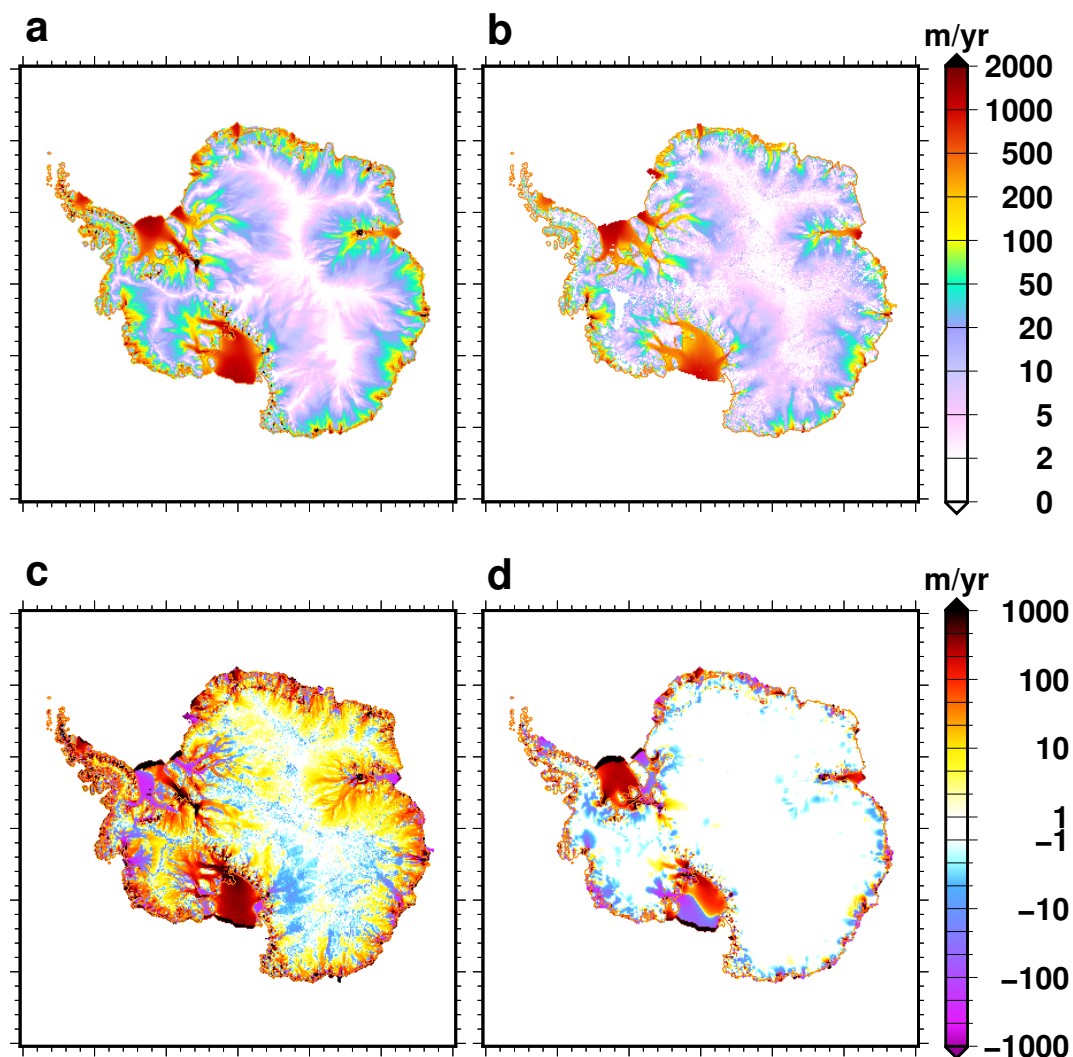

**Figure 3.** Surface velocity magnitude: **(a)** simulated at the end (2011-2015) of the historical experiment *hist*; **(b)** in the observational datasets of Rignot et al. (2011); **(c)** difference between (a) and (b). The surface velocity magnitude change from 2011-2015 to 2096-2100 in the control experiment *ctrl_proj* is shown in **d**. We use a 5 year mean for the simulated velocity to reduce the impact of interannual variability. The range -1 to 1 m yr$^{-1}$ is set to white for velocity difference (**c** and **d**).

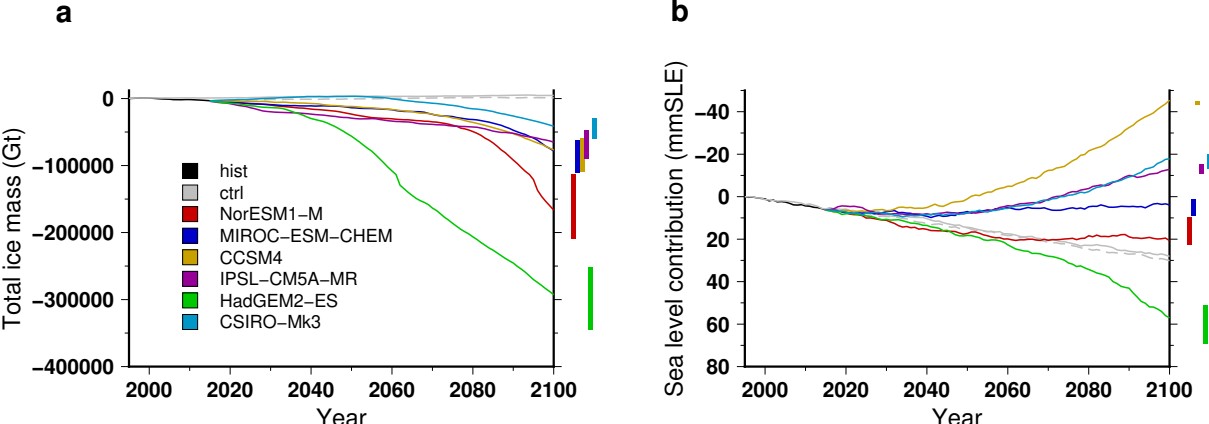

**Figure 4.** Simulated total ice mass change (**a**) and ice volume contributing to sea level rise (**b**) for projections under the different CMIP5 forcings using the RCP8.5 scenario and the medium oceanic sensitivity. The evolutions begin with the historical simulation *hist* (1995-2015) and the control experiments *ctrl* and *ctrl_proj* are depicted in grey (solid and dashed, respectively). For each projection experiment the vertical bar shows the minimal and maximal changes associated with the oceanic forcing sensitivity to temperature change (*low* and *high*).

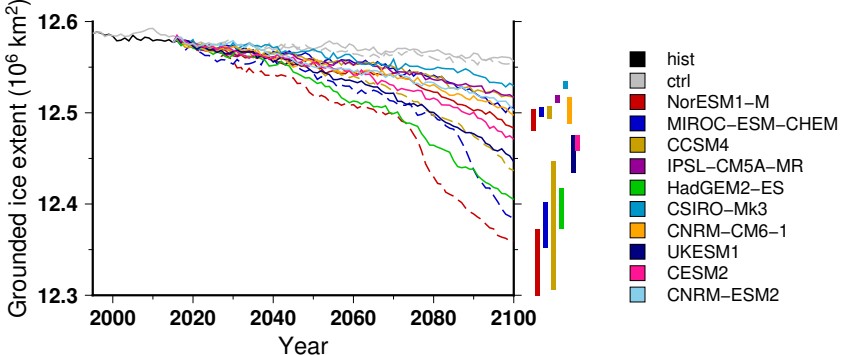

**Figure 5.** Simulated grounded ice extent for projections under the different CMIP5 and CMIP6 climate forcings using the RCP8.5 scenario and SSP585 scenario, respectively. The evolutions begin with the historical simulation *hist* (1995-2015) and the control experiments *ctrl* and *ctrl_proj* are depicted in grey (solid and dashed, respectively). The projection experiments shown in this figure use the medium oceanic sensitivity but for each projection experiment the vertical bar shows the minimal and maximal changes associated with the oceanic forcing sensitivity to temperature change (*low* and *high*). For the projection experiments, the solid lines stand for the experiments that use the sub-shelf melting parametrisation calibrated against all the Antarctic data (*MeantAnt*) while the dashed lines are for the experiments that use a parametrisation calibrated against Pine-Island area data only (*PIGL*).

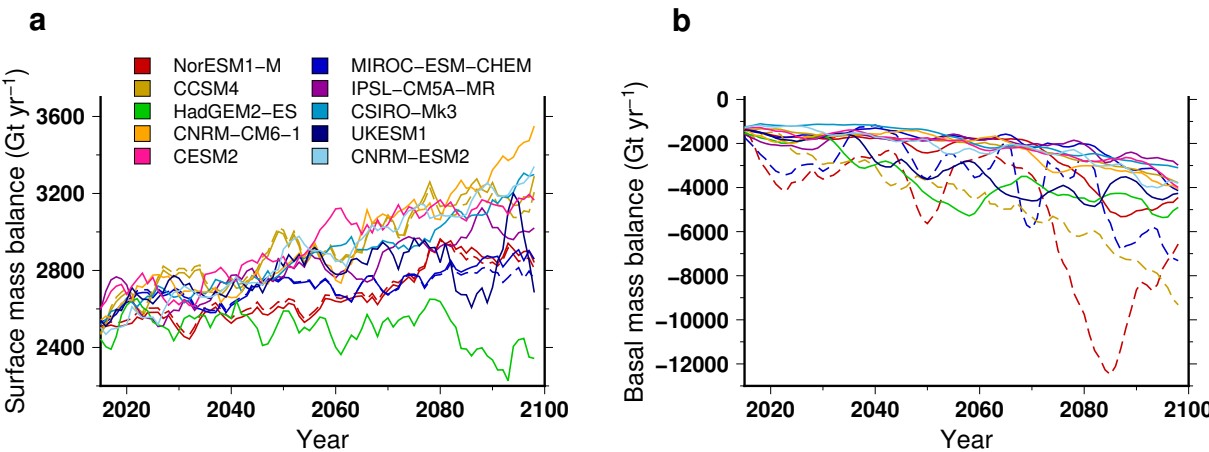

**Figure 6.** Simulated surface mass balance **(a)** and basal mass balance **(b)**, integrated over the ice sheet, for different CMIP5 and CMIP6 climate forcings using the RCP8.5 scenario and SSP585 scenario, respectively. The projection experiments shown in this figure use the medium oceanic sensitivity. The solid lines stand for the experiments that use the sub-shelf melting parametrisation calibrated against all the Antarctic data (*MeanAnt*) while the dashed lines are for the experiments that use a parametrisation calibrated against Pine-Island area data only (*PIGL*). For this figure we use a 5-year running mean in order to smooth the interannual variability.

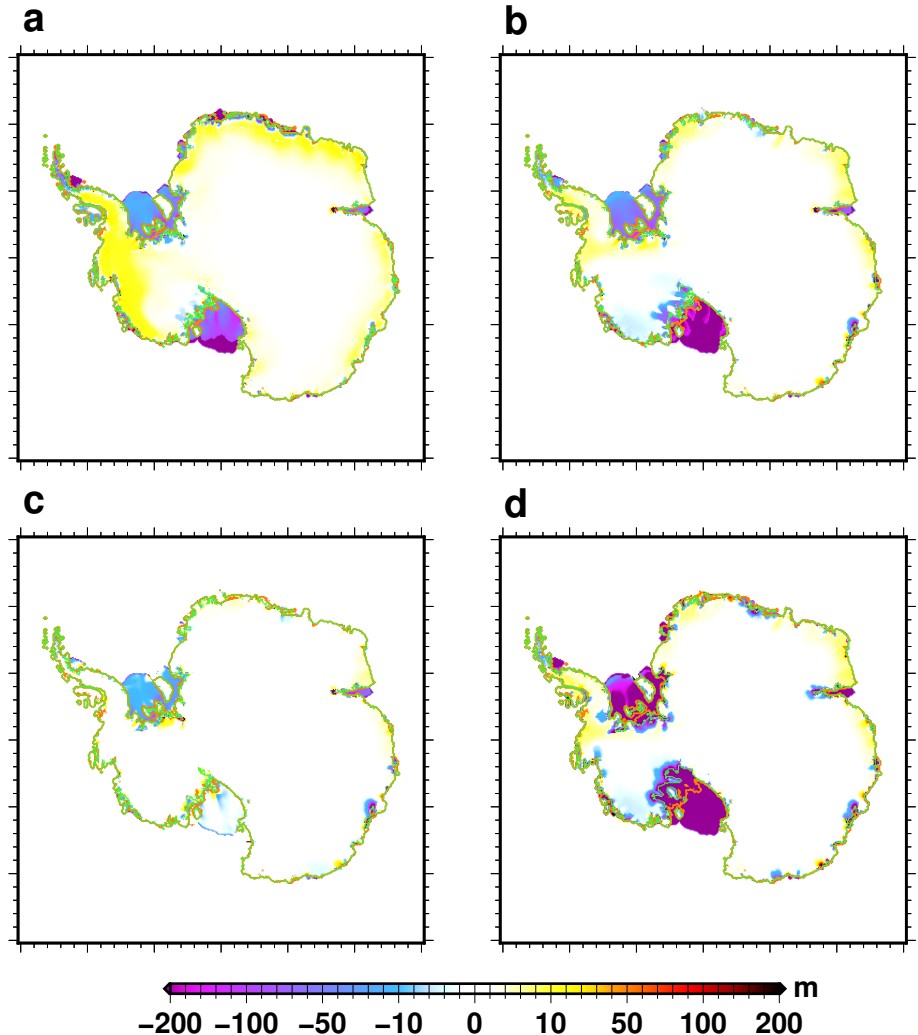

**Figure 7.** Simulated ice thickness change (2100 - 2015) for: **(a)** CESM2 (SSP585); **(b)** NorESM1-M (RCP8.5); **(c)** NorESM1-M (RCP2.6) and; **(d)** NorESM1-M (RCP8.5) *PIGL*. The orange line shows the the present-day grounded line and the light green line represents its simulated position in 2100. The medium oceanic sensitivity has been used here, except for the *PIGL* experiment (d) for which we use the high oceanic sensitivity. The ice thickness change shown here is corrected for the ice thickness change (2100-2015) in the control experiment *ctrl_proj*.

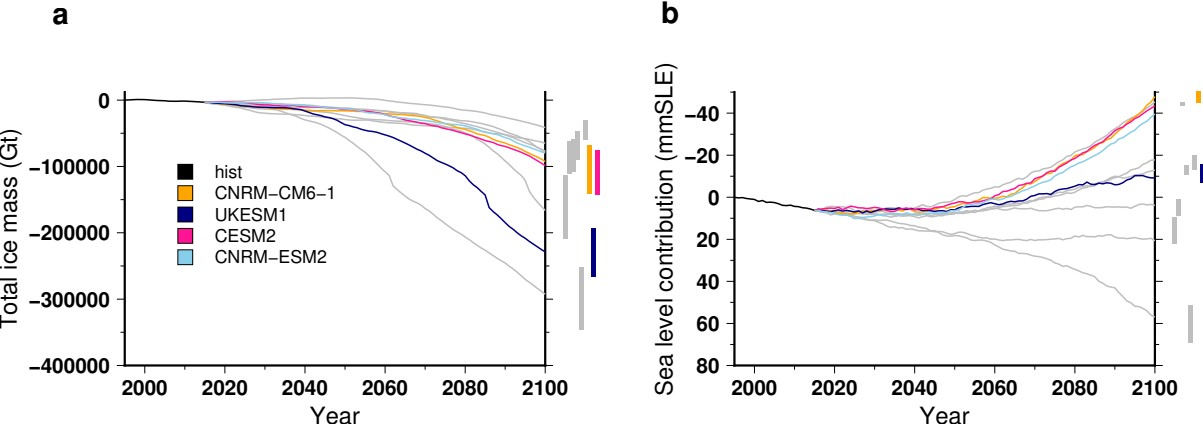

**Figure 8.** Simulated total ice mass change (**a**) and ice volume contributing to sea level rise (**b**) for projections under the different CMIP6 forcings using the SSP585 scenario and the medium oceanic sensitivity. The evolutions begin with the historical simulation *hist* (1995-2015). For each projection experiment the vertical bar shows the minimal and maximal changes associated with the oceanic forcing sensitivity to temperature change (*low* and *high*). The grey lines are the changes under the CMIP5 forcings shown in Fig. 4.

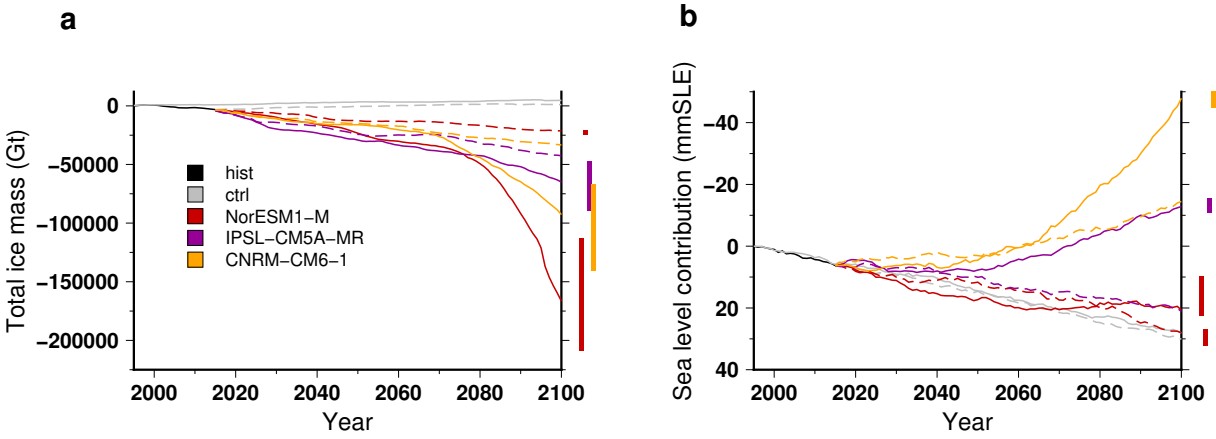

**Figure 9.** Simulated total ice mass change (**a**) and ice volume contributing to sea level rise (**b**) for projections using climate models run under a high (solid lines, RCP8.5 for NorESM1-M and IPSL-CM5A-MR, and SSP585 for CNRM-CM6-1) and a low (dashed lines, RCP2.6 for NorESM1-M and IPSL-CM5A-MR, and SSP126 for CNRM-CM6-1) emission scenario for greenhouse gases with a medium oceanic sensitivity. The evolutions begin with the historical simulation *hist* (1995-2015) and the control experiments *ctrl* and *ctrl_proj* are depicted in grey (solid and dashed, respectively). For each projection experiment the vertical bar shows the minimal and maximal changes associated with the oceanic forcing sensitivity to temperature change (*low* and *high*).

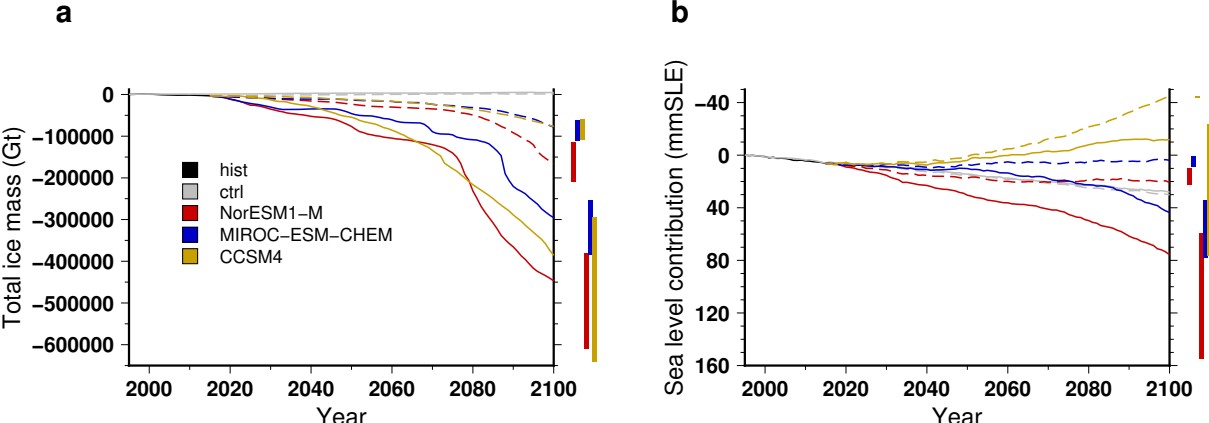

**Figure 10.** Simulated total ice mass change (**a**) and ice volume contributing to sea level rise (**b**) for projections under the different CMIP5 forcings using the RCP8.5 scenario and the medium oceanic sensitivity. For the projections, the solid lines stand for experiments that use a sub-shelf melting rate parametrisation calibrated against the Pine-Island glacier area only (*PIGL*) while the dashed lines stand for experiments that use the sub-shelf melting rate parametrisation calibrated against the Antarctic-wide dataset (*MeanAnt*). The evolutions begin with the historical simulation *hist* (1995-2015) and the control experiments *ctrl* and *ctrl_proj* are depicted in grey (solid and dashed, respectively). For each projection experiment the vertical bar shows the minimal and maximal changes associated with the oceanic forcing sensitivity to temperature change (*low* and *high*).

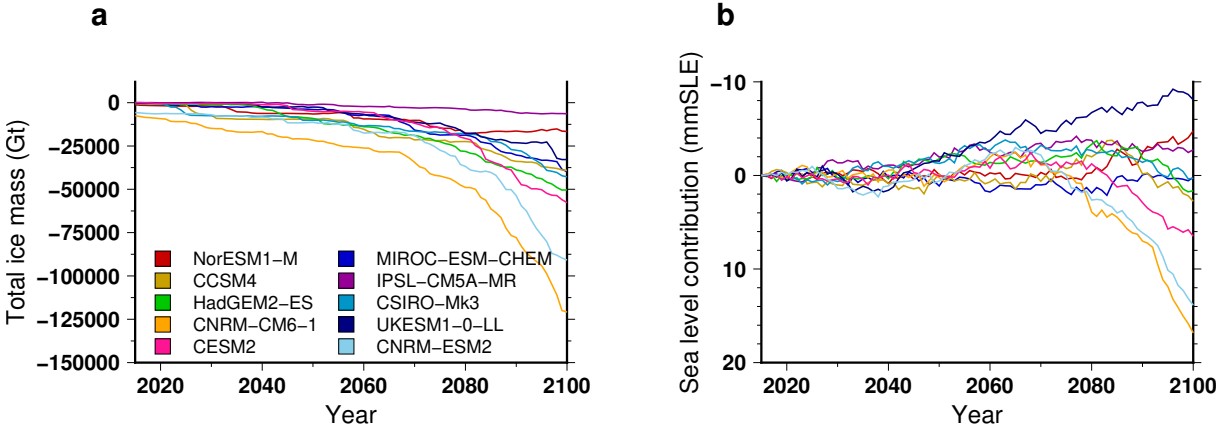

**Figure 11.** Simulated ice volume difference between the shelf collapse scenario and the standard approach for the projections under different CMIP5 and CMIP6 forcings using the RCP8.5 scenario and SSP585 scenario, respectively. The volume change is expressed as: (**a**) total ice mass change and (**b**) ice volume contributing to sea level rise. The projection experiments shown in this figure use the medium oceanic sensitivity.

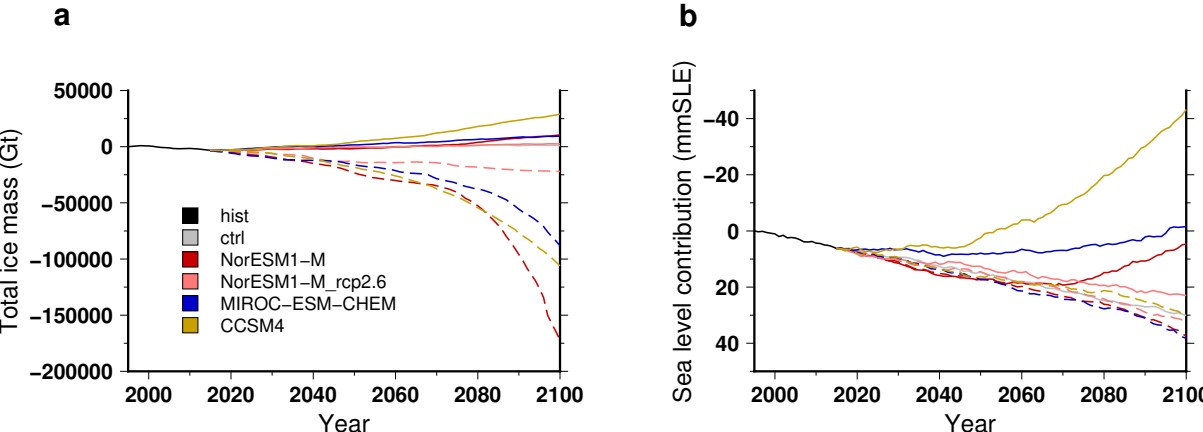

**Figure 12.** Simulated total ice mass change (**a**) and ice volume contributing to sea level rise (**b**) for projections under the different CMIP5 forcings using the RCP8.5 scenario and the medium oceanic sensitivity. For the projections, the solid lines stand for experiments under atmospheric forcing change only (no change in sub-shelf melting rates) while the dashed lines stand for experiments under oceanic forcing change only (no change in surface mass balance nor surface temperature). The evolutions begin with the historical simulation *hist* (1995-2015) and the control experiments *ctrl* and *ctrl_proj* are depicted in grey (solid and dashed, respectively).

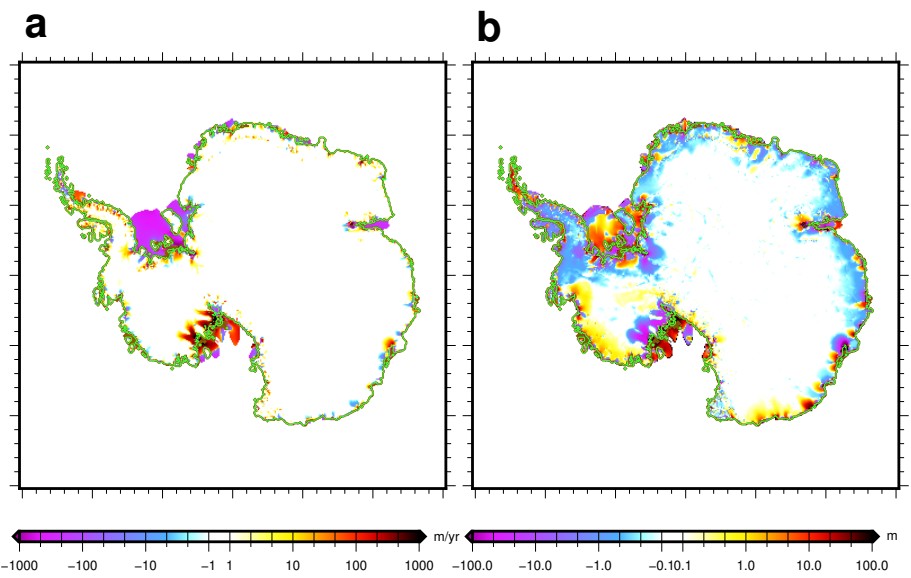

**Figure 13. (a)**: Simulated surface velocity change during the projection run (2096-2100 with respect to 2015-2019) using NorESM1-M forcing under RCP8.5 with a medium oceanic sensitivity. **(b)**: change in the dynamical contribution to ice thickness change in 2100 (see text for definition) for this same experiment. For both panel, we corrected the changes by the ones simulated in the control experiment *ctrl_proj* over the same period. The range -1 to 1 m yr$^{-1}$, respectively -0.1 to 0.1 m, is set to white in **(a)**, respectively **(b)**.

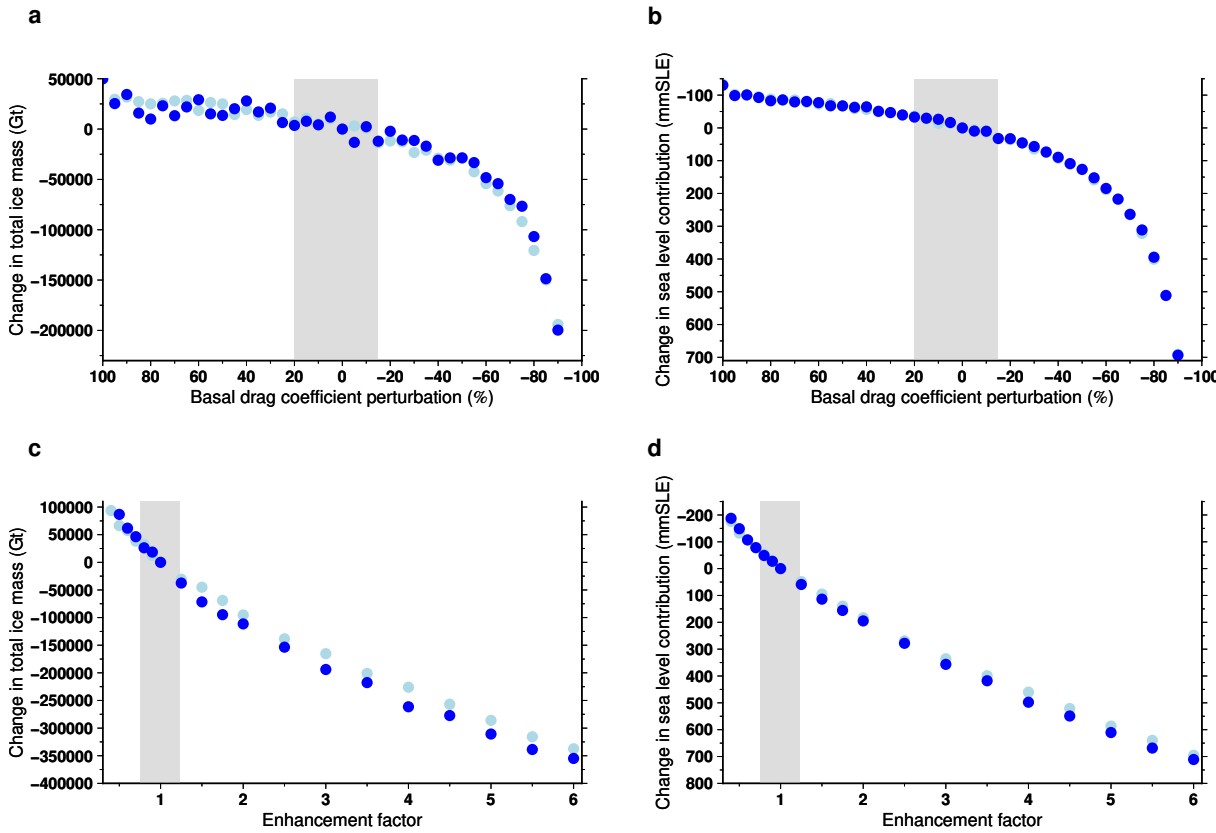

**Figure 14.** Change in ice volume for a modification of the basal drag coefficient (**a** and **b**) and different values of the enhancement factor (**c** and **d**). In this figure, each dot represents the difference in 2100 with respect to the standard experiment (no basal drag coefficient perturbation and enhancement factor at 1). The dark blue dots are projection experiments that use NorESM1-M under RCP8.5 with a medium oceanic sensitivity. The light blue dots are control experiments *ctrl_proj*. Some control experiments can be hidden by the projection experiments if they imply a similar volume change. The perturbations are applied starting at year 2045. The vertical grey band stands for the range of perturbations that produce a 0.15% of total mass change in the perturbed control experiment with respect to the standard control experiment. The difference is expressed in total ice mass (**a** and **c**) and ice volume contributing to sea level rise (**b** and **d**).