# Peer review of "The GRISLI-LSCE contribution to ISMIP6, Part 2: projections of the Antarctic ice sheet evolution by the end of the 21st century"

_The Cryosphere, 2020_

## Referee Comment (RC1) · Johannes Sutter (Referee) · 13 Jul 2020

Quiquet and Dumas present the results of the GRISLI-LSCE contribution to ISMIP6, which is an extensive model intercomparison showing the 21st century evolution of the Antartic Ice Sheet (AIS) under a variety of climate scenarios (Seroussi et al. 2020). I think it is a worthwile excercise to present individual model contributions of ISMIP6 in depth, as due to page limitations and readability the main ISMIP6-paper can only illustrate the general findings of the model intercomparison in broad strokes. Therefore, in my opinion this manuscript is well suited for the scope of The Cryosphere.

[Figure]

An in-depth analysis as the authors attempt here should identify the key features as well as strenghts and weaknesses of the individual model contribution so the reader can appreciate the respective models skills and peculiarities when it comes to projecting the future evolution of the AIS.

The authors present interesting details regarding their model projections and how they differ from or agree with the ISMIP6 ensemble. The paper generally reads well and the figures are of good quality. To make this a valuable addition to the "TC-ISMIP6-canon" I would suggest a number of modifications and extensions mainly with regard to the Results-section as well as some stilistic overhaul to improve the general readability.

I will first list general comments pointing out where certain sections need more substance to elevate this manuscript above a mere documentation of GRISLI ISMIP6-results, followed by specific point by point edits/comments to the text.

1. (section 3.1 Present-day simulated ice sheet)

The authors discuss the modelled present day state of the AIS in detail covering mostly thickness and velocity changes in the different regions of the ice sheet. This gives a nice first impression as to how well GRISLI is capable to reflect the current available observations. If I understand correctly the underlying assumption of the initialisation procedure was to create an ice sheet in equilibrium with the late-20th century mean climate state as opposed to one with ongoing mass loss. If this is correct it could be stressed more, and the consequences of the initialisation for the projection runs (potentially to stable) should be discussed. Furthermore, it would be really interesting to hear the authors opinion on inverting for ice thickness versus inverting for surface velocity. How are the ice sheet's future regional dynamics primed in the projections as a result of the inversion approach? What is the advantage/disadvantage of thickness inversion (e.g. realistic inital geometry/unrealistic flow patterns) in comparison to velocity inversion (e.g. realistic initial ice dynamics/unrealistic surface elevation)? Also, the authors focus a lot on ice shelf thickness and area changes which is important for buttressing

and thus marine ice sheet stability. However, the ice thickness close to the grounding line is probably also an important indicator whether the initial ice sheet configuration is resistant to grounding line retreat or facilitates the latter. In general, it would be nice to have a more explicit discussion as to how the initial state of the ice sheet impacts the projections.

2. (section 3.2.4)

The modelled grounding line response in the Ross and Filchner Ronne sectors seems to be very large for higher sensitivity runs if forced by e.g. NorESM1-M as discussed by the authors and shown in Figure 5 d. I suggest to expand the discussion of this response a little shedding light on the mechanisms and whether this response differs from the ISMIP6 ensemble substantially. Is this solely due to the strong forcing of NorESM1-M in these sectors or is there a model dependence if comparing the different ISMIP6 ensemble members?

3. (section 3.2.5) I think it is an important finding that ice shelf collapse does not seem to have a considerable effect until the year 2100 at least for GRISLI. Section 3.2.5 should be expanded by a discussion as to why this is the case. Did the authors carry out longer projections under ice shelf collapse (e.g. until the year 2300/3000)? Is MISI initialised in certain regions for longer simulation times? Or is the model setup so stable as to not allow MISI (doesn't seem to be the case if looking at Ross Sea grounding line retreat under NorESM RCP8.5 forcing). Is this result similar to the ISMIP6 model ensemble (i.e. do all models show negligible grounding line response to ice shelf collapse untill 2100 CE?), or how does GRISLI differ here? As of now this section is very short and does not really allow for an assessment how sensitive this GRISLI setup is with regard to removal of buttressing force.

4. (Discussion)

Here, the authors discuss a parameter sensitivity study not shown in the results section. Is this on purpose? I would suggest to include a section in the Results and present

the main findings of these experiments there. As for Figure 12 I suggest to include a graphical aid for the reader which delineates what the authors think is a realistic parameter range (e.g. good fit to present day observables). I assume the fringes of the parameter range would generate an ice sheet configuration which are not in agreement with the general present day features of the AIS.

Point by point edits/comments:

general points:

-review your use of "important" (e.g. important acceleration, p1, l18) throughout the manuscript. Important for what? This is very implicit. I know what you mean but the word "important" should be replaced by an explanation of why the change is relevant throughout the manuscript.

-check throughout manuscript "consists in" and change to "consists of" where applicable.

-check your use of "pessimistic" and "optimistic" scenario and replace with e.g. "unmitigated" and "strong mitigation scenario" or alternatively just with the official CMIP abbrev.

-check use of "All together" and replace by e.g. "Overall"

-check use of word "systematic" throughout the manuscript.

-you use the form "on the one hand ... on the other hand" exhaustively, especially in the second half of the manuscript. This is not technically wrong, but it would improve the reading experience if you use other forms to express contrasting things from time to time.

-for sake of readability I suggest to modify occurrences of ice volume changes and write in exponential form (e.g. 3e5 km3 instead of 300000 km3) and provide the sea level equivalent volume change in brackets right after.

Abstract:

p1,l2 this sentence could be changed to: The Antarctic ice sheet's contribution to global sea level rise over the 21st century is of primary societal importance and remains largely uncertain as of yet.

p1,l2-3: ISMIP6 itself suggests a range from negative to positive sea level contribution, while you write "from a few milimetres to more than one metre". This seems inconsistent to me.

p1, l5-6 I suggest to omit: "While in a companion paper we present ..." and shorten the sentence to "Here, we present the GRISLI-LSCE contribution ...".

p1,l8 omit "of sea level equivalent".

p1, l9: suggest to rephrase to "... of the ice shelves resulting in grounding line retreat while increased precipitation partially mitigates or even overcompensates the dynamic ice sheet contribution to global sea level rise."

p1, l12: change "retreats" to "retreat" and check use of retreats throughout the manuscript. p1 l12-13: change to "... in ice sheet models for projections of the Antarctic ice sheet's evolution."

p1, l17 include reference of potential total sea level equivalen ice volume (e.g. $\sim$58.3 m BEDMAP2 or $\sim$57.9 m BEDMACHINE).

p1, l18 rephrase this sentence and include reference, suggestion: "While the ice sheet was probably in a quasi mass-equilibrium in the eighties (citation?), it has since then lost ice at an accelerated pace, reaching a yearly sea level contribution of up to 0.7 mm yr-1 during the last decade (...)"

p1, l21: replace "inexorably" with "irreversible".

p1, l22: change to "While the increase in mass loss is mostly associated with ocean warming, the increased precipitation ..."

p2, l3: "the projected sea level contribution"

p2, l4. please rephrase model formulation, unclear what you mean here. "Overall, the uncertainties related to XY"

p2, l9: "... contribution to ISMIP6-Antarctica in detail, while its ..."

p2, l34: what about shorter timescales such as the one you are looking at here? Has GRISLI taken part in e.g. MISMIP? please elaborate.

p3, l10: "... and the total velocity results from the addition of the ..."

p3, l20: "initialisation procedure consists of ... which aims at determining the geographical ..."

p3, l23: "... under a constant present day ..."

p3, l31-32: please rephrase this sentence, unclear and poor style.

p4, l9 "is derived from" ?

p4,10. "The geothermal heat flux is taken from ..."

p4, l19 "is derived from"

p4, l30: "GCMs". p5, l12: "... as the initial ..."

p5, l15: unclear what you mean by " ... even though the time evolution has no incidence on the forcings."

p5,l15: if I understand correctly you are using annual forcing, so I guess you can omit the specification of the month.

p5, l21: unclear: are they branched of from the historical experiment at 2014?

p5, l27: using different sub-shelf melt rate sensitivities ..

p5, l28 : ...of the sub-shelf melt model calibrated ..."

p6, l1: rephrase " In order to allow for the interpretation of the model response to the forcings, a control experiment, ctrl_proj , has been performed in addition to the ctrl experiment. As in the ctrl experiment ..."

suggest to omit the month as it is not relevant for the simulations

p6,l 19: check your use of "important" and rephrase with an explicit description of what the relevancy is.

p6,l21: the extend

p6,l26: rephrase "location and magnitude" and check use of "important"

p6,l28: rephrase to "Surface velocities of the major tibutaries of the Ross ice shelf (Mercer- and Williams Glacier) and the Filchner-Ronne ice shelves (Foundation Glacier) are largely overestimated (include range here, e.g. up to factor 2 or whatever it is)"

p6, l31. explicitly explain why ross ice shelf is largely over- and Ronne ice shelf under estimated.

p7,l2: which other regions could be affected by the 16 km resolution, i.e. where do you think the grid size plays a dominating role on projections?

p7,9-10: suggest to provide names of ice sheet models which use velocity inversion in ISMIP6 so the reader can compare in Seroussi et al. 2020.

p7,l21 "mass balance uncertainties" - please specify regions with large mass balance uncertainties so the reader can grasp where these are relevant.

p7,l33 "... acceleration of ice volume loss over the course of the century."

p8,l1 as suggested in the general points please write volume changes in exponential form and provide sea level equivalent changes right after in brackets.

p8,l6 "(i.e. above floatation)"

p8,l9 This is not necessarily the whole story as mass gains in grounded ice above sea level could overcompensate mass change in marine ice sheet regions. please elaborate.

p8,l9 : rephrace to "ice shelve volume is shrinking over the course of the century"

p8,l10 : please elaborate "ambivalent"

p8,l11 replace "perpetual" with "constant"

p8,l18 what is the reason for the decresased surface mass balance in HadGEM2-ES? If I plot precip alone over the AIS I get an ∼30% increase. From Seroussi et al. 2020 I gather that surface mass balance anomalies are computed from "changes in precipitation, evaporation, sublimation, and runoff". p8,l18 I assume you mean: basal melting underneath ice shelves is increasing? p8,30: "... wide spread thickening of the grounded ice sheet" p8,31: rephrase to "When using NorESM1-M this thickening is present to a lesser extent and compensated by the thinning that results from the grounding line retreat in some areas (Ross or Totten ice shelves for example)."

p9,l7: "Similar to CMIP5 climate models, the CMIP6 ..."

p9,l10 is this also the case for the ISMIP6 ensemble or a specific feature of your model?

p9,16 "largest" instead of "greater"??

p9,l19: "maintain" what? p9,l24-25: please rephrase these sentences, I know what you mean, but the formulation as it stands is unclear. p9,l29: again, "maintain" what?? p10,l2: "The computation of the sub-shelf melt rate ..." p10,l2 unclear what you mean by "largely derived". I guess the basal melt rate is tuned to observational data. p10,l13: "the NorESM1-M forcing under RCP8.5 ..." p10,l26-28: please completely rephrase this sentence. p10,l30: suggest to omit "Respective" in section header p11,l12-14. please rephrase. p11,l19: suggest to rephrase to : "Modelled grounded ice surface velocity changes are limited with the notable ..." p11,l26: "Another way ... this century ..." p11,l27: replace "different natures" with "different causes" p12,l1: replace "somehow"

with a quantification. p12, l4: unclear what you mean by "in line". Close to ensemble mean? p12,l8 :"(e.g. Bamber et al. ...)" p12,l28: "Such an approach ..." please quantify "much more computationally expensive" p13,l1: "While the atmospheric forcing ... p13,l13: "providing the means to investigate" p13,l20: "partly mitigating or over-compensating the effect of loss of buttressing due to ice shelf melt." p13,l23: "do not drastically change the simulated ice sheet volume ..." p13,l24: "...emission scenarios ..." p13,l24: replace "present" with "exhibits"

General point for the volume figure captions:

you often use the sentence " Simulated ice volume change for the historical experiment hist (1995-2015), the control experiments ctrl (solid grey lines) and ctrl_proj (dashed grey lines) and for the projections using climate models run"

which is a bit bulky and only after that the description of what the panels show follows. For sake of readability I suggest to modify the respective captions so it reads: "Simulated ice volume change and sea level contribution for projections XYZ ..." and in the end include a sentence stating that the plots begin with the historical run and that ctrl and ctrl_proj are depicted in gray (dashed and solid).

Figure 9: How come that for some experiments the AIS sea level contribution is negative for ice shelf collapse in comparison to standard approach? This should be discussed in the results! It seems only those runs which show AIS growth in standard approach show a relative AIS mass loss in the shelf collapse scenarios.

Figure 12: it would help if you indicate the parameter range which produces a "realistic" present day ice sheet with respect to observations for present day forcing, so the reader can identify which parameters are still "OK" to use. Also please remove double brackets e.g. ((a) and (b)) -> (a and b).
* * *

---

## Referee Comment (RC2) · Anonymous Referee #2 · 13 Jul 2020

**Comments to "The GRISLI-LSCE contribution to ISMIP6, Part2: projections of the Antarctic ice sheet evolution by the end of the $21^{st}$ century"**

Aurélien Quiquet and Christophe Dumas

July 13, 2020

**1 General comments**

This paper is based on the Ice Sheet Model Intercomparison project (ISMIP6) on the Antarctic ice sheet. The results of individual ice-sheet model GRISLI are discussed. Apart from the standard experiments described in Seroussi et al., 2020, forcings derived from some CMIP6 model simulations are implemented in this study. Furthermore, experiments with atmospheric forcing only and oceanic forcing only are taken to study their roles separately. Finally, the authors did sensitivity tests on the basal friction coefficient and enhancement factor to address the influence of initial conditions.

Generally, I believe studies based on individual models could be a good complement or further study beyond the intercomparison paper (Seroussi et al., 2020). For example, by implementing different schemes in the single model, uncertainties could be better understood. Though, it's not clear to me what the strong points of this paper are. I have a few concerns about this paper:

- The main results and the induced conclusions are in line with the model intercomparison paper and don't add more information. Therefore I'm not sure why is it important to publish the single model result? There should be more discussion about the regions where the GRISLI model

shows different behavior compared to the mean ISMIP6 model results. (See also specific comments).

- Apart from the standard experiments introduced in Seroussi et al., 2020, the authors added sensitivity experiments on basal drag coefficient and enhancement factor by simply changing the value proportionally. The experiments are only shortly described in the discussion without any contribution to the conclusions. The authors didn't work deeper in this direction of studying the uncertainties from initial conditions.

**2   Specific comments**

Hyphenation should be used between adjective-noun pairs, such as "ice-sheet model", please check through the manuscript.

P1L10: 'sub-shelf basal melt' is a repeated expression.→'sub-ice-shelf melting/melt rates'.

P1L22: 'increased in mass loss'→'acceleration of mass loss'

P2L3: 'ice sheet dynamics'→'ice-sheet dynamics', again, please check through

P2L2:'....remains largely uncertain' need references.

P2L2: delete 'Thus, altogether'?

P2L5: a wide spread in **the prediction/assessment of** the magnitude

P2L9: cite Seroussi et al., 2020

P3L10: I wonder if the total velocity is a weighting function of SIA and SSA as Bueler and Brown, 2009 described or simply added the two velocities? In the later case, the reference should be Winkelmann et al., 2011 (https://doi.org/10.5194/tc-5-715-2011).

P3L24: 'and impose'

P3L28: 'basal drag **coefficient** reduced for ice thickness overestimation', so is the next sentence 'basal drag **coefficient** remains...'

P3L28: 'e.g. basal drag reduced for ice thickness overestimation': how does the coefficient reduce corresponding to the thickness change? The authors should describe the formula clearly, or supply the related references. Similarly, in the sentence of L30, 'The ice thickness mismatch...is used to modify the basal drag coefficient for the next iteration.' How does the ice thickness mismatch modulate the basal drag coefficient?

P3L33: 'Le clec'h et al. (2019)'→'(Le clec'h et al., 2019)'

section 2.2 Model and initialisation: Sensitivity experiments are taken for basal drag coefficient and the enhancement factor, however, the enhancement factor is not introduced in this section. I think it's necessary to describe the parameter, how it influence the stress field and what value do you use in the standard simulations.

P4L8: 'an observational dataset'→'a combination of observational datasets'

P4L25: 'of'→'at'

P4L25–: I suggest to give the non-local quadratic parameterisation formula instead of only refer to the paper. The manuscript heavily discussed the influence of ocean forcing, such as 'sub-shelf melt rates sensitivity to temperature' and the uncertainties related to the 'low','high' and 'medium' methods. However, It's not explained what's the parameter, and what do 'low','medium' and 'high' mean.

P4L28, In the standard experiments, the gamma (sensitivity parameter) has been calibrated to reproduce the total amount of observed sub-ice-shelf melt rate around Antarctica (Rignot et al., 2013).

P4L33, also because there are dense observational data available in Pine Island glacier.

P5L3: Maybe also label the standard calibration as MeanAnt to be consistent with Jourdain et al., 2019.

P5L4: The first sentence need a reference.

P5L7: I didn't find 'SC' used thereafter. Is the sentence needed?

P5L13: 'climate forcings (surface temperature...)' is surface temperature implemented as a forcing?

P5L15: Which forcing is used for the ctrl experiment?

P6L8: delete 'namely GRISLI'?

P6L11: 'These errors are the results of ...' I guess the errors are also from the iterative procedure of initialisation?

P6L15: What do you mean by 'most of the time'?

P6L19  Figure 1: It's not easy for me to tell the yellow color from white. It seems that in the Amundsen sea embayment, there are ∼50 m underestimation of ice thickness in the Getz ice shelf region but ∼50 m overestimateion in Pine Island glacier and Thwaites glacier?

P6L20 'the Filchner-Ronne ice shelf grounding line'→ grounding line of the Filchner-Ronne ice shelf

P6L30: 'The velocity errors for the grounded part...' Why?

P6L31: 'Thus,...' need a more detailed explanation.

P6L7: It's declared in the section 2 that the initialisation method is same with Le clec'h 2019, where the basal drag coefficient is also modulated by velocity. But here you does not have any constraints on the velocities?

P7L20: 'This inconsistency can be due to...' Why? Could you give more specified explanation?

P7L24: '1000 km$^3$' Could you use consistent unit when mentioning the mass change? km$^3$, Gt or sea level equivalent? Right now all of the three units are implemented, making it hard to compare.

P7L26: '...and Filchner-Ronne ice shelves'. Upstream Pine Island, Getz and Totten ice shelves are also quite high? It's not easy to tell from Figure 2d.

P7L32: Using 'MeanAnt' same as Jourdain et al., 2019 instead of 'sub-shelf...dataset' will make it much easier to follow.

P8L27 & Figure 5: 'For both forcings,...' For NorESM1-M the ice-shelf thinning of Totten ice shelf is more pronounced?

P8L31: delete the second 'also'.

P8L33 & Figure 5: This is a very interesting figure which could compare to the Figure 6 of Seroussi et al., 2020. There the mean model result shows an important thinning as well as acceleration in Pine Island, Thwaites and Totten glacier, while the model result for these regions are all quite stable here. However, the explanation here ' This is likely due to the fact that our control experiment tends to produce an ice thickening in this region (Fig. 5b) which tends to stabilise this region, resulting in a smaller sensitivity' is insufficient. Why do you have a thickening trend in the control experiment and why it results in a smaller sensitivity to climate forcings? I noticed from the equations that GRISLI implement linear basal friction law. Brondex et al., 2019 claimed that the Pine Island glacier is sensitive to the sliding laws and an exponent of 8 is suggested for the region. As descriptions of models are listed in Seroussi et al., 2020, I hope the authors can have a more specific discussion.

P9L6: From Figue 6 and Figure 3,4, we can see UkESM1 has more total mass loss compare to NorESM1, and their surface and basal mass balance have similar trend, why NorESM1 has ∼20 mm sea level contribution and UkESM has negative contribution? Is it because of the spatial distribution of forcing?

P9L13: The first sentence can be removed.

P9L16: 'scenarios'

P9L16: 'The model that...' the colors for the three models are really similar.

P9L30: Again, the comparison with the ensemble model results could be interesting.

P11L17: 'NorESM1-M climate forcing'→'NorESM1-M climate forcing under RCP8.5'

P11L18: How does the decrease of surface velocity of ice shelves associated with ice thinning?

P11L31: From Figure 11b, the dynamic contribution in West Antarctica has strong spatial variabilities, e.g. thinning of Siple coast and thickening in Amundsen sea region.

P12L8: '...suggested in other studies' Could you give the numbers from these references?

P12L9: 'One reason for this disagreement...This methodology is thus not suited...' Why this type of initialisation cause the disagreement? And is this the only reason causing disagreements?

P12L19: Enhancement factor appears here for the first time. It should be defined in the methodology. And the author should explain why this parameter is interesting for a sensitivity test.

Figure 12: Explain in the caption or in the text what's the meaning of positive and negative percentages.

P13L8: '...when using the same forcing' I don't think the parameterisations in the open experiments are using the same forcing. At least for PICO, PICOP and Plume, ocean temperature and salinity are used instead of thermal forcing.

P13 section Conclusion: There is not much new information comparing

to the Seroussi et al., 2020 paper.

---

## Referee Comment (RC3) · Fuyuki SAITO (Referee) · 26 Jul 2020

This paper presents a detail of ISMIP6 Antarctic ice experiments using a numerical ice-sheet model GRISLI. In my opinion, it is worthwhile to present detail results of an individual model to participate an intercomparison project, because the corresponding main paper usually focuses on general feature among the participants. I think this paper is fairly well written with some exception below, and can be accepted with minor revision.

[Figure]

There is one relatively major point in the manuscript, which is argued on the experiments shown in Figure 12. In the text the author mentioned that (P12L16): 'A uniform reduction of the basal drag coefficient by 30% leads to a 13000 km3 total volume reduction contributing to about 50 mmSLE in 2100. This means that, with our model, it is unlikely to obtain a significantly different ice volume change for slightly different basal initial conditions.' I do agree the former sentence, but I am not sure what the authors mean in the latter. Is 50 mmSLE insignificant? Or, is 30% change in the basal drag coefficient already too large to be worried about that expected contribution is much smaller than 50 mmSLE? The authors do not provide the inferred basal drag coefficient map in the manuscript. Le clec'h et al. (2019) present the basal drag coefficient map, but for GRISLI Greenland simulation. In this basal drag coefficient map, at least in Greenland ice sheet, the coefficients seem to vary more than a factor thousand. If this factor holds true also for Antarctica, 30% changes in the coefficient may be far smaller than the variation of the coefficients. I appreciate if the author extend this discussion to describe clearer from the experiment design. Moreover, there are not enough information about the sensitivity experiment for the ice enhancement factor, which should be extended.a

Minor points:

P3L9. the abbreviation SSA should be inserted as SIA.

P3L9 and Eq.(2) It is confusing to describe SSA is as a sliding law while a linear till parameterization (2) is used as sliding velocity. Better to explain clearer.

Sect 3.1 and others. There are not a few names of glaciers and the region without explanation. I know that this journal is the Cryosphere and many readers are familiar with such local names, however, I really appreciate if the author show a map of these locations for better understanding of result description.

P7L3, about RMSE of simulated velocity fields. I am interested in the relative rank of RMSE of simulated topography (thickness) by GRISLI. I suspect that the dispersion in

the simulated topography by the participants are smaller than that of the velocity, but I want to know whether GRISLI's errors are both large or only velocity is large among the participants.

P7L14, resemblance of patterns between Fig.1a and b. Why not show a figure of correlation?

P8L12 '... suggesting increased precipitation in the future'. As far as I understand the experiment protocol and the mentioned in the next sentence, changes in simulated ice sheet volume never suggests the precipitation increasing, but it originates from the boundary condition. Please rewrite this part.

Figure 2 and other velocity figures. The range of smallest velocity color (white) is not explicitly written. Or I suspect that it is from +1 m/yr to -1 m/yr, because there are three color boxes between 10 and 100 or 100 and 1000 while only 2 between 1 and 10.

Figure 6 and other evolution figures. Adding numbers of sea-level equivalent height to the ice volume axis (a) will help to compare with (b).

Figure 11b. I do not understand the rule of annotations in the color bar between 0.1 to 10 and -0.1 to +0.1.

SAITO Fuyuki.

---

## Author Comment (AC1) · 20 Oct 2020

**Johannes Sutter**

*Quiquet and Dumas present the results of the GRISLI-LSCE contribution to ISMIP6, which is an extensive model intercomparison showing the 21st century evolution of the Antaric Ice Sheet (AIS) under a variety of climate scenarios (Seroussi et al. 2020). I think it is a worthwile excercise to present individual model contributions of ISMIP6 in depth, as due to page limitations and readability the main ISMIP6-paper can only illustrate the general findings of the model intercomparison in broad strokes. Therefore, in my opinion this manuscript is well suited for the scope of The Cryosphere.*

*An in-depth analysis as the authors attempt here should identify the key features as well as strenghts and weaknesses of the individual model contribution so the reader can appreciate the respective models skills and peculiarities when it comes to projecting the future evolution of the AIS.*

*The authors present interesting details regarding their model projections and how they differ from or agree with the ISMIP6 ensemble. The paper generally reads well and the figures are of good quality. To make this a valuable addition to the "TC-ISMIP6-canon" I would suggest a number of modifications and extensions mainly with regard to the Results-section as well as some stilistic overhaul to improve the general readability.*

Thank you for your thorough reading and your positive appreciation. In the following, we provide a point by point response to your individual comment.

*I will first list general comments pointing out where certain sections need more substance to elevate this manuscript above a mere documentation of GRISLI ISMIP6-results, followed by specific point by point edits/comments to the text.*

*1. (section 3.1 Present-day simulated ice sheet)*
*The authors discuss the modelled present day state of the AIS in detail covering mostly thickness and velocity changes in the different regions of the ice sheet. This gives a nice first impression as to how well GRISLI is capable to reflect the current available observations. If I understand correctly the underlying assumption of the initialisation procedure was to create an ice sheet in equilibrium with the late-20th century mean climate state as opposed to one with ongoing mass loss. If this is correct it could be stressed more, and the consequences of the initialisation for the projection runs (potentially to stable) should be discussed. Furthermore, it would be really interesting to hear the authors opinion on inverting for ice thickness versus inverting for surface velocity. How are the ice sheet's future regional dynamics primed in the projections as a result of the inversion approach? What is the advantage/disadvantage of thickness inversion (e.g. realistic inital geometry/unrealistic flow patterns) in comparison to velocity inversion (e.g. realistic initial ice dynamics/unrealistic surface elevation)? Also, the authors focus a lot on ice shelf thickness and area changes which is important for buttressing and thus marine ice sheet stability. However, the ice thickness close to the grounding line is probably also an important indicator whether the initial ice sheet configuration is resistant to grounding line retreat or facilitates the latter. In general, it would be nice to have a more explicit discussion as to how the initial state of the ice sheet impacts the projections.*

The fact that it produces an ice sheet in quasi-equilibrium with the present-day climate forcing is an important feature of our initialisation procedure and we agree that it should have been discussed more. In the method section, we have now added:
"It should be noted that such initialisation procedure produce an ice sheet in quasi-equilibrium with the late-20th century mean climate state. By construction it does not simulate the accelerated mass loss observed in the last decades (Rignot et al., 2019)."

And in the discussion:
"This methodology is thus not suited to reproduce the recent acceleration in mass loss, particularly important in West Antarctica (Rignot et al., 2019). For example, a simple cumulative value of the observed 2012-2017 loss rate (219 Gt yr$^{-1}$, The IMBIE team, 2018) from 2015 to 2100 will result in an Antarctic ice sheet contribution to sea level rise of 52 cm SLE. This number is much greater than the simulated contribution by GRISLI and more generally, it is much greater than any ISMIP6-Antarctica participating model simulated contribution. This highlights the importance of initial conditions for century scale projections."

About inverting the ice thickness or the ice velocity. Ice thickness (or surface elevation) inversion procedures should, in principle, provide an ice velocity close to the observations (as it should correspond to the balance velocity). Locally, it presents nonetheless some important differences with the observations, differences that can ultimately bias the projections. Compared to the inversion of ice thickness, the inversion of ice velocity can provide a mean to reproduce the recent trend in the observations. To date, only adjoint-based approaches have been followed to inverse the ice velocities. Instead of inverting ice velocities, in future developments of our model we plan to modify the target of the inversion procedure by adding the recent observed ice thickness changes to the observed ice thickness.
We added this in the discussion:
"Assimilation of surface velocities in transient ice sheet simulations are promising methodologies to overcome the limitations inherent to methods that assume steady state (Gillet-Chaulet, 2020). However, they require a complex modelling framework not currently implemented in our ice sheet model. In future developments of our model, we plan to modify the target of the inversion procedure by adding the recent observed ice thickness changes to the observed ice thickness. This would provide a more realistic initial state for the projections."

With respect to the initial state. It is not straightforward to assess the sensitivity of the results to the initial state. We only have one initial state after the initialisation procedure. An alternative could have been to perform multiple inversion procedure for different values of the enhancement factor for example as in Le clec'h et al. (2019). However, the whole initialisation procedure is relatively long to perform and we have done it only for one value of the enhancement factor (=1 which allows for a good performance of the initialisation procedure). This question of the sensitivity of the results to the initial state was part of the objective of the sensitivity experiments in which we apply uniform perturbation to the basal drag coefficient and/or to the enhancement factor. This part has been completely rewritten and extended.

*2. (section 3.2.4)*

*The modelled grounding line response in the Ross and Filchner Ronne sectors seems to be very large for higher sensitivity runs if forced by e.g. NorESM1-M as discussed by the authors and shown in Figure 5 d. I suggest to expand the discussion of this response a little shedding light on the mechanisms and whether this response differs from the ISMIP6 ensemble substantially. Is this solely due to the strong forcing of NorESM1-M in these sectors or is there a model dependence if comparing the different ISMIP6 ensemble members?*

The sector of the Ross ice shelf, together with the sector of the Totten ice shelf, are the two regions that present the largest ice thickness decrease when looking at the average response of the ISMIP6 participating models (Seroussi et al., 2020, Fig. 6a). The Filchner-Ronne ice shelf sector also shows a large ice thickness decrease although this decrease is more localised close to the grounding line. Our results are thus close to the average model response within the ISMIP6 ensemble for these regions. The Pine-Island sector is the one that present the largest standard deviation of ice thickness change amongst participating models (Seroussi et al., 2020, Fig. 6b), with some models that retreat

substantially and other (like GRISLI) that not simulate any substantial changes. Here again, the response of GRISLI lies within the range of ISMIP6 models.

The strength of the oceanic forcing explains why this sector retreat in our simulations. For example, we have a much more limited retreat for IPSL-CM5A-MR since this model has a much smaller thermal forcing in the Ross basin.

We have added this discussion in Sec. 3.2.1 as it is the first time we show the pattern of retreat:
"For the variety of climate forcing used, the Ross and Totten sectors are the ones that most frequently present grounding line retreat and inland thinning. The Filchner-Ronne sector presents also an ice shelf thickness decrease although associated with a limited grounding line retreat. This is consistent with the average response of the ISMIP6 participating models (Fig 6 in Seroussi et al., 2020). The lack of sensitivity of the Pine Island sector is also a feature common to other participating models since the standard deviation of ice thickness change in this area is very high (> 200 m)."

*3. (section 3.2.5) I think it is an important finding that ice shelf collapse does not seem to have a considerable effect until the year 2100 at least for GRISLI. Section 3.2.5 should be expanded by a discussion as to why this is the case. Did the authors carry out longer projections under ice shelf collapse (e.g. until the year 2300/3000)? Is MISI initialised in certain regions for longer simulation times? Or is the model setup so stable as to not allow MISI (doesn't seem to be the case if looking at Ross Sea grounding line retreat under NorESM RCP8.5 forcing). Is this result similar to the ISMIP6 model ensemble (i.e. do all models show negligible grounding line response to ice shelf collapse untill 2100 CE?), or how does GRISLI differ here? As of now this section is very short and does not really allow for an assessment how sensitive this GRISLI setup is with regard to removal of buttressing force.*

The community paper only discussed the impact of ice shelf collapse under CCSM4 (medium oceanic sensitivity). For this forcing, the retreat mask by 2100 has removed ice shelves in the Peninsula and in the Pine Island sector, but affect only very marginally the other ice shelves. First, these sectors are poorly sensitive in GRISLI. Second, CCSM4 is one of the forcing that produces an ice volume evolution mostly driven by the increased precipitation (very limited oceanic forcing with respect to atmospheric forcing, Fig. 5). The ice shelf collapse induces a smaller ice extent (150000 $km^2$ reduction) compared to the standard experiment and, as a result, a smaller integrated surface mass balance. This is why for this forcing the ice shelf collapse is associated to a decrease of the Antarctic contribution with respect to the standard experiment, except in the last 15 years of the century. As a result the impact of the ice shelf collapse is limited with this forcing (2.6 mm SLE) when compared to the average ISMIP6 models (28 mm SLE). However, the standard deviation amongst the ISMIP6 models is also very large, suggesting that some models also show a low sensitivity to this process.

The retreat mask are computed from the outputs of climate models and do not go beyond 2100. Thus, we cannot perform longer experiments. However, we participated with GRISLI to the ABUMIP project in which we quantified the impact of ice buttressing on the simulated ice sheet. To do so, from an equilibrated initial state (different from the one used here), we removed the ice shelves and performed 500 yr simulations. We show in Sun et al. (2020) that GRISLI is able to simulate an important  grounding line retreat of the West Antarctic ice sheet (including the Pine Island sector) when the buttressing force is removed.

We have expanded the discussion on the ice shelf collapse scenarios:
"A greater sensitivity to this process has been reported in Seroussi et al. (2020) (multi-model average of 28 cm SLE in 2100 under the CCSM4 forcing) although associated a wide spread of

response amongst participating models. For most climate models, the retreat masks by 2100 have removed the ice shelves in the Peninsula and in the Pine Island sectors, but affect only very marginally the other ice shelves. In the standard experiments, these sectors show a low sensitivity to the oceanic forcing. In fact, even under the strongest oceanic forcings, GRISLI shows there a limited grounding line retreat. This suggests that the buttressing force is not the reason why the model does not retreat in these sectors. Instead, it is most likely the topographic biases in the initial state that make the model weakly sensitive to the oceanic conditions. Using a different initial state, we have shown in a recent intercomparison exercise (ABUMIP, Sun et al., 2020) that we were able to simulate large grounding line retreat when the buttressing induced by the ice shelves is removed."

*4. (Discussion)*
*Here, the authors discuss a parameter sensitivity study not shown in the results section. Is this on purpose? I would suggest to include a section in the Results and present the main findings of these experiments there. As for Figure 12 I suggest to include a graphical aid for the reader which delineates what the authors think is a realistic parameter range (e.g. good fit to present day observables). I assume the fringes of the parameter range would generate an ice sheet configuration which are not in agreement with the general present day features of the AIS.*

This part has been moved to the results section. It has been considerably extended.

We have also performed an additional set of experiments for which we apply the perturbations to a control experiment. These perturbed control experiments helped us to define the range of acceptable values for the perturbations. To define this range we have selected the perturbed control experiments that show a mass change lower than 0.15% with respect to the standard control experiment (no perturbation). We chose 0.15% of mass change as it represents one tenth of the simulated ice loss in 2100 when using NorESM1-M under RCP8.5 with a medium oceanic sensitivity. The ranges now appear on the figure.

*Point by point edits/comments:*

*general points:*
*-review your use of "important" (e.g. important acceleration, p1, l18) throughout the manuscript. Important for what? This is very implicit. I know what you mean but the word "important" should be replaced by an explanation of why the change is relevant throughout the manuscript.*

Done.

*-check throughout manuscript "consists in" and change to "consists of" where applicable.*

Done.

*-check your use of "pessimistic" and "optimistic" scenario and replace with e.g. "unmitigated" and "strong mitigation scenario" or alternatively just with the official CMIP abbrev.*

Changed to "low/high emission scenario"

*-check use of "All together" and replace by e.g. "Overall"*

Done.

*-check use of word "systematic" throughout the manuscript.*

Done.

*-you use the form "on the one hand ... on the other hand" exhaustively, especially in the second half of the manuscript. This is not technically wrong, but it would improve the reading experience if you use other forms to express contrasting things from time to time.*

Done.

*-for sake of readability I suggest to modify occurrences of ice volume changes and write in exponential form (e.g. 3e5 km3 instead of 300000 km3) and provide the sea level equivalent volume change in brackets right after.*

We have changed the units, the volume change is now expressed as a mass in Gt. We have adopted an exponential form. However, we prefer to keep separating the discussion of total mass change and sea level equivalent as the two numbers show two different evolution.

*Abstract:*

*p1,l2 this sentence could be changed to: The Antarctic ice sheet's contribution to global sea level rise over the 21st century is of primary societal importance and remains largely uncertain as of yet.*

Done

*p1,l2-3: ISMIP6 itself suggests a range from negative to positive sea level contribution, while you write "from a few milimetres to more than one metre". This seems inconsistent to me.*

We were referring to the list of papers cited in P2L6-8 of the original manuscript. However, since the ISMIP6 community paper is now published we should include it. Changed to:
"In particular, in the recent literature, the contribution of the Antarctic ice sheet by 2100 can be negative (sea level fall) by a few centimetres to positive (sea level rise) with some estimates above one metre."

*p1, l5-6 I suggest to omit: "While in a companion paper we present ..." and shorten the sentence to "Here, we present the GRISLI-LSCE contribution ...".*

Done.

*p1,l8 omit "of sea level equivalent".*

We prefer not. It does not make sense to give a volume change in mm if there is no indication of area, does it?

*p1, l9: suggest to rephrase to "... of the ice shelves resulting in grounding line retreat while increased precipitation partially mitigates or even overcompensates the dynamic ice sheet contribution to global sea level rise."*

We have followed your suggestion.

*p1, l12: change "retreats" to "retreat" and check use of retreats throughout the manuscript.*

Done.

*p1 l12-13: change to "... in ice sheet models for projections of the Antarctic ice sheet's evolution."*

Done.

*p1, l17 include reference of potential total sea level equivalen ice volume (e.g. ∼58.3 m BEDMAP2 or ∼57.9 m BEDMACHINE).*

Done:
"Given its size, the Antarctic ice sheet represents the largest single potential contributor in the future, as it represents 58 m of sea level rise if melted completey (Fretwell et al., 2013; Morlighem et al., 2020)"

*p1, l18 rephrase this sentence and include reference, suggestion: "While the ice sheet was probably in a quasi mass-equilibrium in the eighties (citation?), it has since then lost ice at an accelerated pace, reaching a yearly sea level contribution of up to 0.7 mm yr-1 during the last decade (...)"*

Suggestion followed, reference Rignot et al. (2019).

*p1, l21: replace "inexorably" with "irreversible".*

Done.

*p1, l22: change to "While the increase in mass loss is mostly associated with ocean warming, the increased precipitation ..."*

Done.

*p2, l3: "the projected sea level contribution"*

Done.

*p2, l4. please rephrase model formulation, unclear what you mean here. "Overall, the uncertainties related to XY"*

Done.

*p2, l9: "... contribution to ISMIP6-Antarctica in detail, while its …"*

Done.

*p2, l34: what about shorter timescales such as the one you are looking at here? Has GRISLI taken part in e.g. MISMIP? please elaborate.*

No, we have not performed the MISMIP experiments. We have slightly expanded the presentation of GRISLI:
"For century timescales, with the same model version that the one used here, we participated to initMIP-Antarctica (Seroussi et al., 2019), ABUMIP (Sun et al., 2020) and LarMIP (Levermann et al., 2020). Slightly earlier version of the model has been used to simulate the evolution of the Greenland ice sheet until 2100 and 2150 (Peano et al., 2017; Le clec'h et al., 2019a)."

*p3, l10: "... and the total velocity results from the addition of the …"*

Done.

*p3, l20: "initialisation procedure consists of ... which aims at determining the geographical …"*

Done.

*p3, l23: "... under a constant present day …"*

Corrected.

*p3, l31-32: please rephrase this sentence, unclear and poor style.*

This sentence no longer exists. We now provide more information about the initialisation procedure.

*p4, l9 "is derived from" ?*

Changed.

*p4,10. "The geothermal heat flux is taken from …"*

Corrected.

*p4, l19 "is derived from"*

Changed.

*p4, l30: "GCMs".*

Corrected.

*p5, l12: "... as the initial …"*

Corrected.

*p5, l15: unclear what you mean by " ... even though the time evolution has no incidence on the forcings."*

Rephrased:
The *ctrl* experiment starts in January 1995 and ends in Decembre 2100, even though it uses a constant present-day climate forcing (RACMO2.3p2 averaged over 1979-2016).

*p5,l15: if I understand correctly you are using annual forcing, so I guess you can omit the specification of the month.*

Yes, you understand correctly. However, we prefer to keep the specification of the month, for consistency with the community paper and also because we believe it makes it easier to know exactly the length of the different simulations. For example it is not necessarily obvious to know if a 1995-2100 simulation includes the year 2100 or stops after the computations for the year 1999.

*p5, l21: unclear: are they branched of from the historical experiment at 2014?*

Rephrased:
"The different ice sheet projection experiments start in January 2015 and they are all branched from the end of the historical experiment *hist* (Decembre 2014). They end in Decembre 2100 (86 simulated years)."

*p5, l27: using different sub-shelf melt rate sensitivities ..*

Corrected.

*p5, l28 : ...of the sub-shelf melt model calibrated …"*

Corrected.

*p6, l1: rephrase " In order to allow for the interpretation of the model response to the forcings, a control experiment, ctrl_proj , has been performed in addition to the ctrl experiment. As in the ctrl experiment …"*

Done.

*suggest to omit the month as it is not relevant for the simulations*

We prefer to keep the month for consistency with the community paper and also it might provide a more precise idea on the beginning and end of the simulations.

*p6,l 19: check your use of "important" and rephrase with an explicit description of what the relevancy is.*

New version:
"There are large ice thickness underestimations, locally reaching more than 200 metres, in the Getz ice shelf region in the Amundsen sea and upstream the grounding line of the Filchner-Ronne ice shelf."

*p6,l21: the extend*

Rephrased:
"The ice front of the Ross and Filchner-Ronne ice shelves is located about 80 km away from the observations."

*p6,l26: rephrase "location and magnitude" and check use of "important"*

Reformulated:
 "The model generally reproduces the pattern and the magnitude of the observed surface velocities, depicted in Fig. 3b, even if substantial errors remain (Fig. 3c)."

*p6,l28: rephrase to "Surface velocities of the major tibutaries of the Ross ice shelf (Mercer- and Williams Glacier) and the Filchner-Ronne ice shelves (Foundation Glacier) are largely overestimated (include range here, e.g. up to factor 2 or what-ever it is)"*

New version:
"Surface velocities of the major tributaries of the Ross ice shelf (Mercer and Williams glaciers) and Filchner-Ronne ice shelf (Foundation glacier) are largely overestimated (locally up to a factor 4 with errors larger than 1000 m yr$^{-1}$)."

*p6, l31. explicitly explain why ross ice shelf is largely over- and Ronne ice shelf under estimated.*

These biases mostly come from the limitations of our initialisation procedure in which some glaciers can show important velocity errors. We added:
"The velocity errors for the grounded part of the ice sheet mostly explain the velocity errors for the floating ice shelves. Thus, the velocity in the Ross ice shelf is largely overestimated since its tributaries show generally a large ice velocity overestimation. The western part of the Ronne ice shelf shows an opposite behaviour with feeding glaciers showing a velocity underestimation."

*p7,l2: which other regions could be affected by the 16 km resolution, i.e. where do you think the grid size plays a dominating role on projections?*

Added:
"More generally, spatial resolution could explain most of velocity errors in the coastal regions where topography together with spatially variable surface mass balance and sub-shelf melt exert a strong control on simulated velocities."

*p7,9-10: suggest to provide names of ice sheet models which use velocity inversion in ISMIP6 so the reader can compare in Seroussi et al. 2020.*

The initialisation procedure used by the different model is listed in Tab. 3 of Seroussi et al. (2020). Velocity inversion is used in DA and DA+ models. We now provide a few examples:
"[…] ice sheet models that use the velocities in their initialisation procedure (e.g. JPL1_ISSM, UTAS_ElmerIce)."

*p7,l21 "mass balance uncertainties" - please specify regions with large mass balance uncertainties so the reader can grasp where these are relevant.*

There are large mass balance uncertainties all over Antarctica primarily due to the sparse distribution of observational data (e.g. GLACIOCLIM-SAMBA). P7L21 of the original manuscript discusses the biases in the Amery region and it is not intended to discuss Antarctic-wide biases.
We have added some elements for the biases in the Amery region:
"This inconsistency can be due to surface mass balance overestimation in the forcing in this area. This overestimation could be corroborated by the fact that another regional climate model than the one used here simulates a surface mass balance 30% smaller than RACMO2.7 in the Amery region (Agosta et al., 2019)."

*p7,l33 "... acceleration of ice volume loss over the course of the century."*

Corrected.

*p8,l1 as suggested in the general points please write volume changes in exponential form and provide sea level equivalent changes right after in brackets.*

We have kept separated the discussion for the total mass change from the ice mass contributed to sea level rise. We changed the notation though.

Thanks for noticing. Corrected.

*p8,l9 This is not necessarily the whole story as mass gains in grounded ice above sea level could overcompensate mass change in marine ice sheet regions. Please elaborate.*

Locally, this is indeed possible. However, when looking at the spatially integrated numbers, such overcompensation is not reached in the experiments discussed here.

*p8,l9 : rephrace to "ice shelve volume is shrinking over the course of the century"*

Done.

*p8,l10 : please elaborate "ambivalent"*

Reformulated:
"This means that the ice shelves are reducing in volume for all forcings while the grounded ice volume can increase or reduce depending on the forcing used."

*p8,l11 replace "perpetual" with "constant"*

Replaced.

*p8,l18 what is the reason for the decresased surface mass balance in HadGEM2-ES? If I plot precip alone over the AIS I get an ∼30% increase. From Seroussi et al. 2020 I gather that surface mass balance anomalies are computed from "changes in precipitation, evaporation, sublimation, and runoff".*

The HadGEM2-ES surface mass balance anomalies in the future are positive for elevated areas (>2000m) but negative for the coastal areas. The anomalies have been indeed been computed from precipitation minus evaporation minus runoff. We do not have locally the different variables to check but it is possible that the model simulate increased runoff in the future.

*p8,l18 I assume you mean: basal melting underneath ice shelves is increasing?*

Yes, we have followed your suggestion.

*p8,30: "... wide spread thickening of the grounded ice sheet"*

Corrected.
*p8,31: rephrase to "When using NorESM1-M this thickening is present to a lesser extent and compensated by the thinning that results from the grounding line retreat in some areas (Ross or Totten ice shelves for example)."*

Thanks, done.

*p9,l7: "Similar to CMIP5 climate models, the CMIP6 …"*

Changed to: "Similarly to CMIP5 climate models, the CMIP6 [...]"

*p9,l10 is this also the case for the ISMIP6 ensemble or a specific feature of your model?*

The CMIP6 model results have not been studied yet within the ISMIP6 ensemble.

*p9,16 "largest" instead of "greater"??*

Yes, corrected.

We meant: "[…] are able to survive in the course of the century."

*p9,l24-25: please rephrase these sentences, I know what you mean, but the formulation as it stands is unclear.*

Reformulated:
"However, compared to the high emission scenario, the simulated total ice mass evolution using the low emission scenario is closer to the mass evolution of the control experiment. This means that, in this case, the simulated ice sheet changes in the future are dampened with respect to an higher emission scenario."

We meant: "[...]" are able to survive until the end of the century [...]"

Done.

*p10,l2 unclear what you mean by "largely derived". I guess the basal melt rate is tuned to observational data.*

Clarified:
"[…] is a parametrisation tuned to reproduce a combination of observational datasets (Jourdain et al., 2019)."

Corrected.

Rephrased:
"These models show a limited sub-shelf melt (Fig. 5b) and one of the smallest ice mass loss in the future (Fig. 7a). Thus, they produce a large ice shelf extent with respect to the other climate models. CNRM-ESM2 and CNRM-CM6-1 also simulate a pronounced atmospheric warming in the future. This warming is indirectly visible in Fig. 5a since the precipitation increase is primarily driven by the increased temperature. The atmospheric warming together with the large ice shelf extent explain why the CNRM-ESM2 and CNRM-CM6-1 models show the largest mass loss resulting from ice shelf collapse."

*p10,l30: suggest to omit "Respective" in section header*

Done.

Rephrased:
"Conversely, the total ice mass change (Fig. 11a) mostly reflects the mass loss from the ice shelves which respond primarily to the oceanic forcing. The ice shelf mass loss in the OO experiments can be large with an important acceleration in the last 20 years of the century. This late response might be a reason why the volume above floatation is not drastically different from the control experiment in the OO experiments."

*p11,l19: suggest to rephrase to : "Modelled grounded ice surface velocity changes are limited with the notable ..."*

Suggestion followed.

*p11,l26: "Another way ... this century …"*

Corrected.

*p11,l27: replace "different natures" with "different causes"*

Done.

*p12,l1: replace "somehow" with a quantification.*

The quantification is given at the beginning of the sentence ("a few centimetres"). We have replaced "somehow" by "slightly":
"In East Antarctica there is a widespread very small (a few centimetres) negative dynamical contribution to ice thickness change (ice thinning) that slightly moderates the ice thickening due to increased precipitation."

*p12, l4: unclear what you mean by "in line". Close to ensemble mean?*

Yes, rephrased:
"Although close to the ensemble mean of the ice sheet models participating in [...]"

*p12,l8 :"(e.g. Bamber et al. ...)"*

Corrected.

*p12,l28: "Such an approach ..." please quantify "much more computationally expensive"*

Added:
"Such approach is much more computationally expensive since it requires multiple regional climate model simulations. For example, the MAR regional climate model (Agosta et al., 2019) requires about 15 days to compute 100 years (C. Agosta, personal communication). That is why this approach has been discarded so far for the Antarctic ice sheet where [...]"

*p13,l1: "While the atmospheric forcing ...*

Done.

*p13,l13: "providing the means to investigate"*

Corrected.

*p13,l20: "partly mitigating or over-compensating the effect of loss of buttressing due to ice shelf melt."*

Done.

*p13,l23: "do not drastically change the simulated ice sheet volume ..."*

Corrected.

*p13,l24: "...emission scenarios..."*

Corrected.

*p13,l24: replace "present" with "exhibits"*

Done.

*General point for the volume figure captions:*

*you often use the sentence " Simulated ice volume change for the historical experiment hist (1995-2015), the control experiments ctrl (solid grey lines) and ctrl_proj (dashed grey lines) and for the projections using climate models run"*
*which is a bit bulky and only after that the description of what the panels show follows.*
*For sake of readability I suggest to modify the respective captions so it reads: "Simulated ice volume change and sea level contribution for projections XYZ ..." and in the end include a sentence stating that the plots begin with the historical run and that ctrl and ctrl_proj are depicted in gray (dashed and solid).*

Thanks for the suggestion. We have followed your advice.

*Figure 9: How come that for some experiments the AIS sea level contribution is negative for ice shelf collapse in comparison to standard approach? This should be discussed in the results! It seems only those runs which show AIS growth in standard approach show a relative AIS mass loss in the shelf collapse scenarios.*

It is true that it is counter-intuitive. For some climate forcings, the ice shelf collapse scenario produces a local thickening in the vicinity of some grounded line. This is mostly related to local non-linearities. We have added the following:
"The impact of the ice shelf collapse scenario on the sea level contribution ranges from -8 to +17 mm SLE. This range is much smaller than the range of the simulated sea level contribution for the different climate models (-50 to 70 mm SLE). Surprisingly, for some models, the ice shelf collapse scenario contributes negatively to the sea level contribution (e.g. UKESM1-0-LL). This is most probably due to local non-linearities of grounding line dynamics. However this effect is limited to small changes in the grounded volume."

CCSM4 shows a negative contribution to future sea level rise with an evolution very similar to CNRM-CM6-1, CNRM-ESM2 and CESM2. However, while the shelf collapse scenario increases the contribution to sea level rise in 2100 for CNRM-CM6-1, CNRM-ESM2 and CESM2, it has a negligible impact on CCSM4. It is therefore not obvious to draw a general conclusion.

*Figure 12: it would help if you indicate the parameter range which produces a "realistic" present day ice sheet with respect to observations for present day forcing, so the reader can identify which parameters are still "OK" to use. Also please remove double brackets e.g. ((a) and (b)) -> (a and b).*

We have added a vertical grey band for the acceptable range (volume change in perturbed control with respect to the standard control lower than 0.15%). Double brackets removed.

**References**

Oppenheimer, M., Glavovic, B. C., Hinkel, J., van De Wal, R. S. W., Magnan, A. K., Abd-Elgawad, A., Cai, R., CifuentesJara, M., DeConto, R. M., Ghosh, T., Hay, J., Isla, F., Marzeion, B., Meyssignac, B., and Sebesvari, Z.: Sea Level Rise and Implications for Low-Lying Islands, Coasts and Communities, in: IPCC Special Report on the Ocean and Cryosphere in a Changing Climate, edited by: H.-O. Pörtner, D. C. R., V. Masson-Delmotte, P. Zhai, M. Tignor, E. Poloczanska, K. Mintenbeck, A. Alegría, M. Nicolai, A. Okem, J. Petzold, B. Rama, N. M. Weyer, 2019.

Rignot, E., Mouginot, J., Scheuchl, B., van den Broeke, M., van Wessem, M. J., and Morlighem, M.: Four decades of Antarctic Ice Sheet mass balance from 1979–2017, Proceedings of the National Academy of Sciences, 116, 1095–1103, doi:10.1073/pnas.1812883116, 2019.

Seroussi, H., Nowicki, S., Payne, A. J., Goelzer, H., Lipscomb, W. H., Abe-Ouchi, A., Agosta, C., Albrecht, T., Asay-Davis, X., Barthel, A., Calov, R., Cullather, R., Dumas, C., Galton-Fenzi, B. K., Gladstone, R., Golledge, N. R., Gregory, J. M., Greve, R., Hattermann, T., Hoffman, M. J., Humbert, A., Huybrechts, P., Jourdain, N. C., Kleiner, T., Larour, E., Leguy, G. R., Lowry, D. P., Little, C. M., Morlighem, M., Pattyn, F., Pelle, T., Price, S. F., Quiquet, A., Reese, R., Schlegel, N.-J., Shepherd, A., Simon, E., Smith, R. S., Straneo, F., Sun, S., Trusel, L. D., Van Breedam, J., van de Wal, R. S. W., Winkelmann, R., Zhao, C., Zhang, T., and Zwinger, T.: ISMIP6 Antarctica: a multi-model ensemble of the Antarctic ice sheet evolution over the 21st century, The Cryosphere, 14, 3033–3070, doi:https://doi.org/10.5194/tc-14-3033-2020, 2020.

Sun, S., Pattyn, F., Simon, E. G., Albrecht, T., Cornford, S., Calov, R., Dumas, C., Gillet-Chaulet, F., Goelzer, H., Golledge, N. R., Greve, R., Hoffman, M. J., Humbert, A., Kazmierczak, E., Kleiner, T., Leguy, G. R., Lipscomb, W. H., Martin, D., Morlighem, M., Nowicki, S., Pollard, D., Price, S., Quiquet, A., Seroussi, H., Schlemm, T., Sutter, J., Wal, R. S. W. v. d., Winkelmann, R., and Zhang, T.: Antarctic ice sheet response to sudden and sustained ice-shelf collapse (ABUMIP), Journal of Glaciology, pp. 1–14, doi:10.1017/jog.2020.67, publisher: Cambridge University Press, 2020.

---

## Author Comment (AC2) · 20 Oct 2020

**1 General comments**

This paper is based on the Ice Sheet Model Intercomparison project (ISMIP6) on the Antarctic ice sheet. The results of individual ice-sheet model GRISLI are discussed. Apart from the standard experiments described in Seroussi et al., 2020, forcings derived from some CMIP6 model simulations are implemented in this study. Furthermore, experiments with atmospheric forcing only and oceanic forcing only are taken to study their roles separately. Finally, the authors did sensitivity tests on the basal friction coefficient and enhancement factor to address the influence of initial conditions.

Thank you for your careful reading. In the following we provide a point by point response to your comments.

Generally, I believe studies based on individual models could be a good complement or further study beyond the intercomparison paper (Seroussi et al., 2020). For example, by implementing different schemes in the single model, uncertainties could be better understood. Though, it's not clear to me what the strong points of this paper are. I have a few concerns about this paper:

• The main results and the induced conclusions are in line with the model intercomparison paper and don't add more information. Therefore I'm not sure why is it important to publish the single model result? There should be more discussion about the regions where the GRISLI model shows different behavior compared to the mean ISMIP6 model results. (See also specific comments).

We acknowledge that the conclusions of our paper are not drastically different from the one in Seroussi et al. (2020). This is in part due to the fact that GRISLI shows a model response close to the mean of the ensemble of ISMIP6 participating models. However, beyond the general conclusion, we think that papers that show an individual group contribution to a large intercomparison exercise have three main advantages:

- Documentation. The model response to the forcings is clearly reported in a single model paper while this information can be buried in the community paper. The documentation of a specific model response is important to analyse any further studies that use this model.

- Climate forcing uncertainty quantification. The community paper is best suited for a quantification of the sensitivity to the choice of the ice sheet model while the sensitivity to the climate forcing is better shown for individual model.

- Model bias description. Very limited information on individual model biases is given in the community paper. Such issues are more extensively discussed in a single model paper.

We have added these ideas in the introduction section:

"The analysis of a single model response to the different forcing scenarios presents some important added value with respect to the community paper of Seroussi et al. (2020). First, single model paper allows for a documentation of a specific model response to the forcings while this information can be buried in the community paper given the large material to cover. Second, the community paper is best suited for a quantification of the sensitivity of the projections to the choice of the ice sheet model. The sensitivity to the climate forcing is better shown for individual ice sheet model. Third, single model paper can provide a more complete information of model biases."

• Apart from the standard experiments introduced in Seroussi et al., 2020, the authors added sensitivity experiments on basal drag coefficient and enhancement factor by simply changing the value proportionally. The experiments are only shortly described in the discussion without any

**contribution to the conclusions. The authors didn't work deeper in this direction of studying the uncertainties from initial conditions.**

Seroussi et al. (2020) only describe the results for Tier 1 and Tier 2. These experiments are limited to CMIP5 climate forcing and only cover a subset of the sensitivities to RCP/SSP scenarios, subshelf melt calibration and shelf collapse scenarios. Excluding the "open" experiments (which are mutually exclusive with the "standard" experiments), Seroussi et al. (2020) discuss 12 different experiments. Here we discuss 60 experiments from which new features not discussed in Seroussi et al. (2020), such as results for the CMIP6 forcing and atmospheric and oceanic only experiments.

**We added this in the introduction:**

"Thanks to a relatively low computational cost, we performed the full list of experiments of ISMIP6 described in Nowicki et al. (2020), where Seroussi et al. (2020) only cover a subset of these experiments."

In addition, we also performed 38x2 additional experiments with a perturbed basal drag coefficient and 17x2 additional experiments varying the flow enhancement factor. We have completely rewritten the description of the results of these perturbed experiments. This part has been also largely extended and we think it brings valuable information on the choice of the initial ice sheet state. However, in order to fully explore the sensitivity of the model results to the initial state we would have needed different initial state. For example, we could have run multiple initialisation procedures for different values of the flow enhancement factor as in Le clec'h (2019). However, the whole initialisation procedure is relatively long to perform and we have done it only for one value of the enhancement factor (=1 which allows for a good performance of the initialisation procedure).

**2 Specific comments**

**Hyphenation should be used between adjective-noun pairs, such as "ice-sheet model", please check through the manuscript.**

Hopefully corrected, unsure for some cases. There is a great variety of spellings in the published literature, even among native speakers. Eventually, the Copernicus language editing service will be able to correct the mistakes that we might have overlooked.

P1L10: 'sub-shelf basal melt' is a repeated expression.  $\rightarrow$  'sub-ice-shelf melting/melt rates'.

Corrected.

P1L22: 'increased in mass loss'  $\rightarrow$  'acceleration of mass loss'

Corrected

*P2L3: 'ice sheet dynamics'*  $\rightarrow$  *'ice-sheet dynamics', again, please check through*

Done.

P2L2: '....remains largely uncertain' need references.

We though that the 9 references in the following sentence should suffice. We have nonetheless added a reference to the special report of the IPCC (Oppenheimer et al., 2019).

P2L2: delete 'Thus, altogether'?

Changed for "Overall".

*P2L5: a wide spread in the prediction/assessment of the magnitude*

Corrected.

P2L9: cite Seroussi et al., 2020

Done.

P3L10: I wonder if the total velocity is a weighting function of SIA and SSA as Bueler and Brown, 2009 described or simply added the two velocities? In the later case, the reference should be Winkelmann et al., 2011 (https://doi.org/10.5194/tc-5-715-2011).

It is a simple addition indeed. We have added the reference to Winkelmann et al. (2011).

P3L24: 'and impose'

Done.

P3L28: 'basal drag coefficient reduced for ice thickness overestimation', so is the next sentence 'basal drag coefficient remains...'

**Corrected.**

P3L28: 'e.g. basal drag reduced for ice thickness overestimation': how does the coefficient reduce corresponding to the thickness change? The authors should describe the formula clearly, or supply the related references. Similarly, in the sentence of L30, 'The ice thickness mismatch...is used to modify the basal drag coefficient for the next iteration.' How does the ice thickness mismatch modulate the basal drag coefficient?

We now give more information on the manuscript, including the two main equations.

P3L33: 'Le clec'h et al. (2019)' → '(Le clec'h et al., 2019)'

Corrected.

section 2.2 Model and initialisation: Sensitivity experiments are taken for basal drag coefficient and the enhancement factor, however, the enhancement factor is not introduced in this section. I think it's necessary to describe the parameter, how it influence the stress field and what value do you use in the standard simulations.

We have added the following:

"As in most large-scale ice sheet models, GRISLI uses a flow enhancement factor to artificially account for ice anisotropy (Quiquet et al., 2018). In the model, we specify the value of this enhancement factor for the SIA velocity and we use a fixed ratio to determine its smaller SSA counterpart. For the experiments presented here (except in Sec. 3.2.7), we use a flow enhancement factor of 1 (no SIA enhancement) and a ratio close to 1 for the SSA (1.2:1)."

P4L8: 'an observational dataset'  $\rightarrow$  'a combination of observational datasets'

Done.

P4L25: 'of'  $\rightarrow$  'at'

Corrected.

P4L25–: I suggest to give the non-local quadratic parameterisation formula instead of only refer to the paper. The manuscript heavily discussed the influence of ocean forcing, such as 'sub-shelf melt rates sensitivity to temperature' and the uncertainties related to the 'low', 'high' and 'medium' methods. However, It's not explained what's the parameter, and what do 'low', 'medium' and 'high' mean.

Ok, we have substantially rewritten the description of the sub-ice-shelf melt parametrisation, providing the equation and more details on the calibration.

P4L28, In the standard experiments, the gamma (sensitivity parameter) has been calibrated to reproduce the total amount of observed sub-ice-shelf melt rate around Antarctica (Rignot et al., 2013).

Thanks, we have clarified this.

P4L33, also because there are dense observational data available in Pine Island glacier.

Added.

*P5L3:* Maybe also label the standard calibration as MeanAnt to be consistent with Jourdain et al., 2019.

Done.

P5L4: The first sentence need a reference.

Added, Scambos et al. (2009).

P5L7: I didn't find 'SC' used thereafter. Is the sentence needed?

Right, removed.

P5L13: 'climate forcings (surface temperature...)' is surface temperature implemented as a forcing?

Yes it is. The model is thermo-mechanically coupled and surface temperature is a boundary condition for the temperature diffusion.

P5L15: Which forcing is used for the ctrl experiment?

RACMO2.3p2 averaged over 1979-2016. Precision added in the manuscript.

P6L8: delete 'namely GRISLI'?

Done.

*P6L11: 'These errors are the results of ...'I guess the errors are also from the iterative procedure of initialisation?*

Not directly since we restart from the observations for the ice thickness: the errors are simply due to the drift during the 65-yr relaxation. Of course the chosen map for basal drag coefficients will drive this drift.

*P6L15: What do you mean by 'most of the time' ?*

Simplified:

"The differences over the East Antarctic plateau are smaller than a few metres but increases towards the ice margins or in the vicinity of major ice streams (e.g. Amery ice shelf tributaries)"

P6L19 Figure 1: It's not easy for me to tell the yellow color from white. It seems that in the Amundsen sea embayment, there are  $\sim$ 50 m underestimation of ice thickness in the Getz ice shelf region but  $\sim$ 50 m overestimateion in Pine Island glacier and Thwaites glacier?

We have changed the colour palette, hopefully it is now clearer. Yes, we have ~50 m overestimation in Pine Island and Thwaites glacier regions. The error in the Getz ice shelf region is slighly larger, reaching 200 m locally.

P6L20 'the Filchner-Ronne ice shelf grounding line'  $\rightarrow$  grounding line of the Filchner-Ronne ice shelf

Done.

*P6L30: 'The velocity errors for the grounded part...' Why?*

For a large upstream flux, mass conservation will favour a large downstream flux as well.

P6L31: 'Thus,...' need a more detailed explanation.

We give slightly more information:

"Thus, the velocity in the Ross ice shelf is largely overestimated since its tributaries show generally a large ice velocity overestimation. The western part of the Ronne ice shelf shows an opposite behaviour with feeding glaciers showing a velocity underestimation."

P6L7: It's declared in the section 2 that the initialisation method is same with Le clec'h 2019, where the basal drag coefficient is also modulated by velocity. But here you does not have any constraints on the velocities?

It is exactly the same methodology as in Le clec'h et al. (2019). There is no constraints on velocities in Le clec'h et al. (2019).

*P7L20: 'This inconsistency can be due to...' Why? Could you give more specified explanation?*

The control experiment should ideally have no drift in ice thickness as it is based on the assumption that the ice sheet is at equilibrium. In the Amery region we have an ice thicknening in the control, suggesting that the ice velocity should be higher. However, the ice velocity is already too high when compared to the observations. This inconsistency can be the results of a too high mass balance in the climate forcing.

We have rephrased this idea:

"The ice thickening during the control experiment could suggest an underestimation of the ice velocity, i.e. underestimation of the ice export, which seems in contradiction to the overestimation of the simulated ice velocity with respect to the observations. This inconsistency can be due to surface mass balance overestimation in the forcing in this area."

*P7L24*: '1000 km3 ' Could you use consistent unit when mentioning the mass change? km3 , Gt or sea level equivalent? Right now all of the three units are implemented, making it hard to compare.

Sea level equivalent and total ice mass (or volume) can not be used interchangeably, since only a fraction of the total mass (or volume) contributes to sea level rise. However, we have switch for total mass (in Gt) change instead of total volume (in km3). In doing so, the mass balance and the total mass are given in a comparable unit.

*P7L26: '...and Filchner-Ronne ice shelves'. Upstream Pine Island, Getz and Totten ice shelves are also quite high? It's not easy to tell from Figure 2d.*

The colour palette in these maps have been changed. We have added: "Although more localised, the changes in Pine Island, Getz and Totten areas can be larger than one hundred metres per year."

P7L32: Using 'MeanAnt' same as Jourdain et al., 2019 instead of 'sub- shelf...dataset' will make it much easier to follow.

We have added the label *MeanAnt*.

*P8L27* & Figure 5: 'For both forcings,...' For NorESM1-M the ice-shelf thinning of Totten ice shelf is more pronounced?

For both forcing this ice shelf has disappeared by 2100.

P8L31: delete the second 'also'.

Done.

P8L33 & Figure 5: This is a very interesting figure which could compare to the Figure 6 of Seroussi et al., 2020. There the mean model result shows an important thinning as well as acceleration in Pine Island, Thwaites and Totten glacier, while the model result for these regions are all quite stable here. However, the explanation here 'This is likely due to the fact that our control experiment tends to produce an ice thickening in this region (Fig. 5b) which tends to stabilise this region, resulting in a smaller sensitivity' is insufficient. Why do you have a thickening trend in the control experiment and why it results in a smaller sensitivity to climate forcings? I noticed from the equations that GRISLI implement linear basal friction law. Brondex et al., 2019 claimed that the Pine Island glacier is sensitive to the sliding laws and an exponent of 8 is suggested for the region. As descriptions of models are listed in Seroussi et al., 2020, I hope the authors can have a more specific discussion.

**We have added the following elements:**

"Our model does not simulate substantial changes in the Pine Island glacier area. In this region, there is a thickening of the ice sheet during the control experiment (Fig. 2b) with underestimated surface velocities (Fig. 3c). These biases can be due to the inferred basal drag coefficient during the initialisation procedure that leads to an underestimation of the velocities. The linear friction law implemented in our model can also result in an underestimation of the velocity (Brondex et al.,

2019). Finally, the biases can also be the result of the complex topographic setting that might not be well captured at 16 km. The underestimated ice sheet velocity at the grounding line in this area, together with the thickening bias, result in a small sensitivity to oceanic warming. However, for other intercomparison exercices we have shown that our model is able to produce a grounding line retreat in this area (Sun et al., 2020).

For the variety of climate forcing used, the Ross and Totten sectors are the ones that most frequently present grounding line retreat and inland thinning. The Filchner-Ronne sector presents also an ice shelf thickness decrease although associated with a limited grounding line retreat. This is consistent with the average response of the ISMIP6 participating models (Fig 6 in Seroussi et al., 2020). The lack of sensitivity of the Pine Island sector is also a feature common to other participating models since the standard deviation of ice thickness change in this area is very high (>~200 m)."

P9L6: From Figue 6 and Figure 3,4, we can see UkESM1 has more total mass loss compare to NorESM1, and their surface and basal mass balance have similar trend, why NorESM1 has ~20 mm sea level contribution and UkESM has negative contribution? Is it because of the spatial distribution of forcing?

Until 2080, UKESM1 shows a larger surface mass balance than NorESM1 (about 200 Gt/yr difference early in the century) and a smaller basal mass balance than NorESM1 (reaching about 1000 Gt/yr difference circa 2050). With this, it is somehow expected that UKESM1 shows the largest total mass loss (ice shelf melting) but the smallest sea level contribution to sea level rise when compared to NorESM1. The spatial distribution of the forcing can explain partly the difference (NorESM1 has only a larger SMB than UKESM1 at the margins) but it is most probably of the second order in this case.

P9L13: The first sentence can be removed.

Done.

P9L16: 'scenarios'

Corrected.

P9L16: 'The model that...' the colors for the three models are really similar.

We have changed the colours used for the different models.

P9L30: Again, the comparison with the ensemble model results could be interesting.

No map showing the impact of the scenario is shown in Seroussi et al. (2020). However, the response in term of ice sheet contribution to sea level rise is discussed for two climate forcing. We have added the following in the manuscript:

"In Seroussi et al. (2020), two climate forcings (NorESM1-M and IPSL-CM5A-MR) were evaluated for both the RCP2.6 and the RCP8.5. The simulated contribution to sea level rise in the ISMIP6 ensemble is very similar to the GRISLI response: no change in grounded ice mass for NorESM1-M but an increase in grounded ice mass for IPSL-CM5A-MR under RCP8.5 with respect to RCP2.6. CNRM-CM6-1 shows a response similar to the one of the IPSL-CM5A-MR since the grounded ice mass is increasing under the SSP585 with respect to the SSP126."

*P11L17: 'NorESM1-M climate forcing' → 'NorESM1-M climate forcing under RCP8.5'*

Done.

**P11L18: How does the decrease of surface velocity of ice shelves associated with ice thinning?**

It is more the thinning that induces a velocity reduction (the SSA velocity is positively correlated with the ice thickness). Locally, the ice thinning in the vicinity of the grounding line can induce a smaller ice flux feeding the ice shelf.

P11L31: From Figure 11b, the dynamic contribution in West Antarctica has strong spatial variabilities, e.g. thinning of Siple coast and thickening in Amundsen sea region.

Yes it does although the positive contribution in the Amundsen sea region are generally small (less than 10 metres). We have added the following:

"In West Antarctica, the dynamical contribution has a strong spatial variability. It can reach up to more than 50 metres decrease in ice thickness and [...]"

P12L8: '...suggested in other studies' Could you give the numbers from these references?

We now refer to the IPCC special report here:

"A relatively moderate Antarctic ice sheet contribution to future sea level rise by 2100 has also been suggested in other studies since the IPCC special report on the ocean and cryosphere in a changing climate (Oppenheimer et al., 2019) reported a range from 30 to 280 mm SLE (RCP8.5)."

P12L9: 'One reason for this disagreement...This methodology is thus not suited...' Why this type of initialisation cause the disagreement? And is this the only reason causing disagreements?

By construction, a methodology that produces an ice sheet in equilibrium under present-day climate cannot, at the same time, reproduce the recent observed acceleration of mass loss. Other source of uncertainties are listed in the discussion and, notably, the sub-ice-shelf melt model since the largest simulated mass loss among ISMIP6 participating model is systematically obtained with ice sheet models that use their own sub-ice-shelf melt model (open experiments) instead of the standard ISMIP6 approach.

P12L19: Enhancement factor appears here for the first time. It should be defined in the methodology. And the author should explain why this parameter is interesting for a sensitivity test.

The enhancement factors are describe in the methodology section now.

Figure 12: Explain in the caption or in the text what's the meaning of positive and negative percentages.

We have added the following:

"The perturbation starts from +100% (i.e. a doubling of the base value) to -90% (i.e. a reduction to 10% of the base value)."

P13L8: '...when using the same forcing' I don't think the parameterisations in the open experiments are using the same forcing. At least for PICO, PICOP and Plume, ocean temperature and salinity are used instead of thermal forcing.

The ISMIP6 thermal forcing is also computed from the ocean temperature and salinity. But it is true that in the standard experiments, the ice sheet models do not use directly the temperature and salinity as forcing. We have changed the sentence to:

"when using forcings elaborated from the same climate model realisations."

P13 section Conclusion: There is not much new information comparing to the Seroussi et al., 2020 paper.

GRISLI is not an outsider within the ISMIP6 ensemble and as a result the numbers given in the conclusion are not out of the ISMIP6 range. We have added few elements regarding the additional sensitivity tests we performed:

"Finally, with additional simple sensitivity tests we have shown that the simulated ice sheet contribution to sea level rise by 2100 could be largely affected by changes in ice-sheet mechanical properties such as basal dragging. Given the weak understanding on such processes, they could also represent a large source of uncertainty."

**References**

Oppenheimer, M., Glavovic, B. C., Hinkel, J., van De Wal, R. S. W., Magnan, A. K., Abd-Elgawad, A., Cai, R., CifuentesJara, M., DeConto, R. M., Ghosh, T., Hay, J., Isla, F., Marzeion, B., Meyssignac, B., and Sebesvari, Z.: Sea Level Rise and Implications for Low-Lying Islands, Coasts and Communities, in: IPCC Special Report on the Ocean and Cryosphere in a Changing Climate, edited by: H.-O. Pörtner, D. C. R., V. Masson-Delmotte, P. Zhai, M. Tignor, E. Poloczanska, K. Mintenbeck, A. Alegría, M. Nicolai, A. Okem, J. Petzold, B. Rama, N. M. Weyer, 2019.

Seroussi, H., Nowicki, S., Payne, A. J., Goelzer, H., Lipscomb, W. H., Abe-Ouchi, A., Agosta, C., Albrecht, T., Asay-Davis, X., Barthel, A., Calov, R., Cullather, R., Dumas, C., Galton-Fenzi, B. K., Gladstone, R., Golledge, N. R., Gregory, J. M., Greve, R., Hattermann, T., Hoffman, M. J., Humbert, A., Huybrechts, P., Jourdain, N. C., Kleiner, T., Larour, E., Leguy, G. R., Lowry, D. P., Little, C. M., Morlighem, M., Pattyn, F., Pelle, T., Price, S. F., Quiquet, A., Reese, R., Schlegel, N.-J., Shepherd, A., Simon, E., Smith, R. S., Straneo, F., Sun, S., Trusel, L. D., Van Breedam, J., van de Wal, R. S. W., Winkelmann, R., Zhao, C., Zhang, T., and Zwinger, T.: ISMIP6 Antarctica: a multi-model ensemble of the Antarctic ice sheet evolution over the 21st century, The Cryosphere, 14, 3033–3070, doi:https://doi.org/10.5194/tc-14-3033-2020, 2020.

Sun, S., Pattyn, F., Simon, E. G., Albrecht, T., Cornford, S., Calov, R., Dumas, C., Gillet-Chaulet, F., Goelzer, H., Golledge, N. R., Greve, R., Hoffman, M. J., Humbert, A., Kazmierczak, E., Kleiner, T., Leguy, G. R., Lipscomb, W. H., Martin, D., Morlighem, M., Nowicki, S., Pollard, D., Price, S., Quiquet, A., Seroussi, H., Schlemm, T., Sutter, J., Wal, R. S. W. v. d., Winkelmann, R., and Zhang, T.: Antarctic ice sheet response to sudden and sustained ice-shelf collapse (ABUMIP), Journal of Glaciology, pp. 1–14, doi:10.1017/jog.2020.67, publisher: Cambridge University Press, 2020.

---

## Author Comment (AC3) · 20 Oct 2020

**Fuyuki Saito**

*This paper presents a detail of ISMIP6 Antarctic ice experiments using a numerical ice-sheet model GRISLI. In my opinion, it is worthwhile to present detail results of an individual model to participate an intercomparison project, because the corresponding main paper usually focuses on general feature among the participants. I think this paper is fairly well written with some exception below, and can be accepted with minor revision.*

Thank you for your positive evaluation. We address your concerns in the following.

*There is one relatively major point in the manuscript, which is argued on the experiments shown in Figure 12. In the text the author mentioned that (P12L16): 'A uniform reduction of the basal drag coefficient by 30% leads to a 13000 km3 total volume reduction contributing to about 50 mmSLE in 2100. This means that, with our model, it is unlikely to obtain a significantly different ice volume change for slightly different basal initial conditions.' I do agree the former sentence, but I am not sure what the authors mean in the latter. Is 50 mmSLE insignificant? Or, is 30% change in the basal drag coefficient already too large to be worried about that expected contribution is much smaller than 50 mmSLE? The authors do not provide the inferred basal drag coefficient map in the manuscript. Le clec'h et al. (2019) present the basal drag coefficient map, but for GRISLI Greenland simulation. In this basal drag coefficient map, at least in Greenland ice sheet, the coefficients seem to vary more than a factor thousand. If this factor holds true also for Antarctica, 30% changes in the coefficient may be far smaller than the variation of the coefficients. I appreciate if the author extend this discussion to describe clearer from the experiment design. Moreover, there are not enough information about the sensitivity experiment for the ice enhancement factor, which should be extended.*

We agree. As in Le clec'h et al. (2019), the basal drag coefficient in Antarctica shows a very high spatial variability. This coefficient can vary from ~1 to $10^5$ Pa yr m$^{-1}$. However, in practice a value above $10^3$ Pa yr m$^{-1}$ produce very limited sliding velocities. Also, the absolute value of the basal drag coefficient has none or a limited impact in the interior of the ice sheet, where the SSA velocity is small anyway, but is very important in the coastal regions, where the ice streams are located.

Our approach is very simple as we applied a uniform perturbation. It allows for an artificial speed-up of the ice streams but it is not suited to investigate realistic changes that could occur in the future. For example, for a realistic ice sheet, it can be envisioned that a grounded point switches from a state where it slowly flows to an ice stream state. In this case, in the model, it means that the basal drag coefficient switches from a value greater than 1000 to lower than 100 Pa yr m$^{-1}$. To test such phenomenon in the model we could apply a random noise in the basal drag coefficient with much larger perturbation than the one we used here.

The problem is that we only have one map for the basal drag coefficients, being the one obtained after the initialisation procedure. Ideally we should have tested alternative maps. However, if such alternative maps not resulting from our inversion were used, it would have resulted in unwanted drift in the control simulation. The use of these maps would be difficult to justify.

We added a number of new simulations in order to get an idea of what range of values for the uniform perturbation is acceptable. We now perform new ctrl_proj simulations in which the basal drag coefficient (and the enhancement factor) is perturbed in the same way as the NorESM1-M projection shown in the initial version of the manuscript. We computed the volume drift of these perturbed ctrl_proj experiments and compared it to the volume drift in the standard ctrl_proj experiment. We consider that a 0.15% difference between the standard ctrl_proj experiment and the perturbed one is acceptable. We chose 0.15% of the volume difference since it corresponds to 10%

of the change in volume simulated in 2100 using NorESM1-M under RCP8.5 (medium oceanic sensitivity meanAnt).

This discussion has been moved from the discussion section to the results section. It has been largely rephrased and extended as well.

*Minor points:*

*P3L9. the abbreviation SSA should be inserted as SIA.*

Added.

*P3L9 and Eq.(2) It is confusing to describe SSA is as a sliding law while a linear till parameterization (2) is used as sliding velocity. Better to explain clearer.*

We simply rephrased to:
"For temperate regions, we assume a linear basal friction (Weertman, 1957):"

*Sect 3.1 and others. There are not a few names of glaciers and the region without explanation. I know that this journal is the Cryosphere and many readers are familiar with such local names, however, I really appreciate if the author show a map of these locations for better understanding of result description.*

We have added a map as Fig. 1.

*P7L3, about RMSE of simulated velocity fields. I am interested in the relative rank of RMSE of simulated topography (thickness) by GRISLI. I suspect that the dispersion in the simulated topography by the participants are smaller than that of the velocity, but I want to know whether GRISLI's errors are both large or only velocity is large among the participants.*

The RMSE of simulated ice thickness was given in P6 L22-23 of the original manuscript. It is about 120 metres and it is the 5th lowest in the ISMIP6 ensemble (21 models). As a result, compared to the other participating models, only the velocity error is large for GRISLI.

*P7L14, resemblance of patterns between Fig.1a and b. Why not show a figure of correlation?*

Not sure what you meant. The spatial correlation between two 2D variables is a scalar right? We computed a Pearson correlation of 0.24 between the two variables shown in Fig 1a and b. We have added this value in the main manuscript. This relatively low value can be explained by the noisy signal of the ice thickness difference between the end of the historical experiment and the observations.
To visualise this correlation we could plot the thickness difference at the end of the *ctrl_proj* as a function of the thickness difference at the end of the historical experiment *hist*. We show this figure in this response (Fig. R1). We are unsure if this will bring additional value? If yes, we would be happy to add such a figure in the paper.

[Figure]

**Figure R1:** Ice thickness difference at the end of the control experiment ctrl_proj with respect to the end of the historical experiment as a function of the ice thickness difference at the end of the historical experiment with respect to observations (Frettwell et al., 2013). The red line represents the linear regression with a correlation value at 0.24.

*P8L12 '... suggesting increased precipitation in the future'. As far as I understand the experiment protocol and the mentioned in the next sentence, changes in simulated ice sheet volume never suggests the precipitation increasing, but it originates from the boundary condition. Please rewrite this part.*

Modified for:
"In addition, except under the HadGEM2-ES forcing, the Antarctic contribution to global sea level rise is always smaller than under the control experiment under constant present-day forcing. This suggests that the climate forcing computed from the GCMs in the future leads to a larger integrated total mass balance compared to our reference present-day mass balance. In fact, most GCMs simulate an increase in precipitation in Antarctica related to the projected warming."

*Figure 2 and other velocity figures. The range of smallest velocity color (white) is not explicitly written. Or I suspect that it is from +1 m/yr to -1 m/yr, because there are three color boxes between 10 and 100 or 100 and 1000 while only 2 between 1 and 10.*

We have changed the colour scale. The range -1 to 1 m/yr is white. This is now specified in Fig. 3 (former Fig. 2).

*Figure 6 and other evolution figures. Adding numbers of sea-level equivalent height to the ice volume axis (a) will help to compare with (b).*

We are not sure what you want us to do here. To express the volume shown in (a) in sea-level equivalent instead of in km$^3$ (or Gt)? We prefer not to do so as it might appear confusing for the reader to express in cm SLE a volume change that is not contributing to sea level rise. What we could do instead is to express all the volume changes in Gt instead of using the sea-level equivalent. However, we think that most people are interested in the sea-level equivalent so we prefer to use

this unit. In order to facilitate the comparison of the two panels, we have added the conversion factor (1 mm SLE = 372 Gt) in the figure captions when applicable.

*Figure 11b. I do not understand the rule of annotations in the color bar between 0.1 to 10 and -0.1 to +0.1.*

We have changed the colour scale. The range from -0.1 to 0.1 m is white. This is now specified in the caption.

---

## Referee Report (RR1)

Revision No. 2 of : The GRISLI-LSCE contribution to ISMIP6, Part 2: projections of the Antarctic ice sheet evolution by the end of the 21st century.

I thank the authors for their revision of their manuscript. The authors adressed the majority of my previous comments but some parts of the manuscript still lack a more rigid discussion of aspects such as grounding line sensitivity and the differentiation between precipitation and surface mass balance. There also remain some stylistic and spelling issues but they are very minor.

Please refer to the attached commented-pdf for my remaining comments/remarks.

[revised manuscript text omitted]

---

## Author Response (AR2)

**Editor review**

Thank you again for taking the time to serve as Editor on our manuscript.

I would like to see the revision of the manuscript based on the comments by the referees, with the response point by point. Especially I expect to see more discussion, which is more than describing the single model, after seeing the ISMIP6 now published. Please use this opportunity to expand the discussion on the response of the grounding line to the oceanic forcing, for example.

We apologise since we did not provide information about grounding line implementation in GRISLI in previous versions of the manuscript. We have added the following:

"Since the model is generally used at a coarse resolution (greater than 5 km), we use an analytical formulation of the flux at the grounding line following either Schoof (2007) or Tsai et al. (2015). The sub-grid position of the grounding line is estimated with a linear extrapolation of the floatation criteria. From this sub-grid position of the grounding line, the ice flux from Schoof (2007) or Tsai et al. (2015) is extrapolated to the neighbouring velocity grid points. More details on this implementation is provided in Quiquet et al. (2018). Using a 40 km grid resolution the model was able to reproduce glacial-interglacial grounding line migration in agreement with geological data (Quiquet et al., 2018). A 16 km version was also used to assess the importance of buttressing for grounding line stability in the ABUMIP intercomparison exercise (Sun et al., 2020), where GRISLI shows an important grounding line retreat although amongst the lowest within the other participating models. Here, we use the analytical flux of Schoof (2007) at the grounding line."

We have also added a discussion on the grounding line sensitivity in GRISLI compared to other models. The first paragraph of the discussion section now reads:

"Amongst the different experiments, the largest contribution by 2100 is 150 mm SLE (NorESM1-M PIGL with a high oceanic sensitivity) while most experiments produce a contribution no greater than 80 mm SLE. Thus, it appears that the contribution of the Antarctic ice sheet to global sea level rise simulated by GRISLI is relatively limited. Since ISMIP6-Antarctica was a large intercomparison exercise that involved 13 research groups and 21 model versions, it is useful to compare these numbers with the ISMIP6-Antarctica ensemble. For this ensemble, using a medium oceanic sensitivity, HadGEM2-ES produces the largest mass loss with an ensemble mean of 96 mm SLE and CCSM4 produces the largest mass gain with an ensemble mean of -37 mm SLE. Although GRISLI does not stand up as an outlier within the ISMIP6 ensemble, it shows a more limited sea level contribution with 58 mm SLE for HadGEM2-ES and -45 mm SLE for CCSM4. This could suggest a moderate sensitivity of the grounding line migration in response to the oceanic forcing when compared to the other ice sheet models. However, it is important to note that some outliers are largely influencing the ISMIP6-Antarctica ensemble mean towards higher contributions. In particular, some ice sheet models that do not use the standard ISMIP6 approach to compute subshelf melting (open experiments) produce much higher ice sheet mass loss. Notably, for NorESM1-M (RCP8.5 medium oceanic sensitivity), ULB\_FETISH32\_open, ULB\_FETISH16\_open, VUB\_PISM\_open and NCAR\_CISM\_open simulate a 2100 mass loss ranging from 72 mm SLE to 163 mm SLE where all the other models show an ensemble mean close to 0 mm SLE. In addition, when models use both the *standard* and the *open* approach to compute the sub-shelf melting, the open approach tends to produce much higher mass loss (NCAR\_CISM, UCIJPL\_ISSM, ULB\_FETISH32, ULB\_FETISH16). Thus, it seems that the consideration of how the different groups have implemented this process is crucial to understand the multi-model spread. When we consider only the models that use the *standard* approach, GRISLI shows a mass loss much closer to the ensemble mean. However, it is not excluded that GRISLI shows a relatively low oceanic sensitivity. It is for example unable to simulate any substantial grounding line retreat in the Pine Island glacier area for the different climate scenarios tested here, even though this could be linked to initialisation model biases that induce an ice thickening in this area in the control experiment. Also, in the ABUMIP intercomparison exercise (Sun et al., 2020), GRISLI shows one of the lowest grounding line retreat due to the loss of buttressing (3rd lowest grounded ice loss in 500 years with respect to the control, out of 15 participating models). Sun et al. (2020) suggested that plastic friction laws produce greater grounding line sensitivity than linear friction law as the one used here. This was also suggested by Brondex et al. (2019). A foreseen improvement of our ice sheet model will be the implementation of various friction laws to better assess the sensitivity of grounding line dynamics to this process."

**One question from my side on the model description is how you treat the calving of this model.**

We have added some information in the model description section:

"Calving is based on a simple threshold criterion: the ice thickness at the front reaching a minimal value is automatically calved if the upstream flux is not sufficient to maintain a thickness above this critical threshold. The minimal ice thickness is set to 200 m in the experiments presented here."

**Johannes Sutter review**

I thank the authors for their revision of their manuscript. The authors addressed the majority of my previous comments but some parts of the manuscript still lack a more rigid discussion of aspects such as grounding line sensitivity and the differentiation between precipitation and surface mass balance. There also remain some stylistic and spelling issues but they are very minor.

*Please refer to the attached commented-pdf for my remaining comments/remarks.*

Thank you for these valuable comments. We provide a point by point response to each one of them in the following. A track-change version of our manuscript is uploaded with this response.

**I* recommend to:**

1. sharpen the discussion of the grounding line sensitivity in the model used here, and provide an assessment whether it under- or overestimates the response to 21st century oceanic forcing.

We added a discussion on the sensitivity of the grounding line in GRISLI. This new paragraph can be found in this document as a response to the first Editor comment.

2. distinguish between surface mass balance and precipitation throughout the manuscript or even better only use surface mass balance.

Thank you for pointing this issue out. We now mostly mention the surface mass balance and we only use "precipitation" for very specific discussions.

P1 L19 please specify what kind of (if any) grounding line parameterisation is used here? The MISMIP intercomparison excercise has shown that coarse resolution models (i.e. several km or > 10 km) do not exhibit reversible grounding lines. Feldmann et al. have shown e.g. for PISM, that a subgrid interpolation of the grounding line position can alleviate that allowing for reversibe grounding lines even at coarse resolution. I do not know whether this is discussed in Quiquet et al. but it would be certainly important to quickly not here what is done with respect to grounding line parameterisations.

We realise that we did not provide any information on the grounding line formulation in GRISLI in our manuscript. We have added more information in the model description section with this respect:

"Since the model is generally used at a coarse resolution (greater than 5 km), we use an analytical formulation of the flux at the grounding line following either Schoof (2007) or Tsai et al. (2015). The sub-grid position of the grounding line is estimated with a linear extrapolation of the floatation criteria. From this sub-grid position of the grounding line, the ice flux from Schoof (2007) or Tsai et al. (2015) is extrapolated to the neighbouring velocity grid points. More details on this implementation is provided in Quiquet et al. (2018). Using a 40 km grid resolution the model was able to reproduce glacial-interglacial grounding line migration in agreement with geological data (Quiquet et al., 2018). A 16 km version was also used to assess the importance of buttressing for grounding line stability in the ABUMIP intercomparison exercise (Sun et al., 2020), where GRISLI shows an important grounding line retreat although amongst the lowest within the other participating models. Here, we use the analytical flux of Schoof (2007) at the grounding line."

*P9* L14-15 either there is a typo in the mass loss specs or the adjectives are swapped. I assume there is a typo as HadGEM2-ES is one of the models showing the most warming.

There was a typo (a zero missing), HadGEM2-ES produces effectively the largest mass loss  $(300 \times 10^3 \text{ Gt in } 2100)$ . It has been corrected.

**P9 L22 actually, it could be worth including a plot of the change in grounded ice area as this gives an indication of integrated grounding line changes**

Thank you for your suggestion. You have included such a plot and added an additional point in the result section:

"An other way to show this is to investigate the grounding line migration in the course of the century. In Fig. 5 we show the grounded ice extent evolution which is an integrated indicator of grounding line migration. For all the projection experiments, the grounded ice extent is always smaller than in the control experiment, and this extent decreases in the course of the century. Thus, even for models that produce an important grounded ice volume increase in the future (e.g. CCSM4), the grounded ice extent is decreasing. This can be only explained by an increase in surface mass balance over the grounded area."

P9 L22-24 this is a very important result. To make your figures consistent with the figures in the main ISMIP6 publication, I strongly suggest to change figure 3 and all figures showing the sea level equivalent mass change so they are showing the change relative to the ctrl\_simulation. I know that you plot the ctrl and historical simulation as well but it is easier for the reader if she/he does not have to subtract it in her/his head.

We understand your point. However, we would prefer to keep the plots that show the absolute changes and not the anomalies relative to the *ctrl\_proj* simulation, for several reasons:

- the absolute changes reflect what is really happening in the model while the anomalies have no real physical meaning. It is true that it is sometimes convenient to show the anomalies in order to remove the potential drift of the model but in doing so we assume that the drift is preserved between the control and the projections (the effect of the projections being simply added to the drift). Since it is an assumption that is not necessarily verified, we prefer to show the absolute changes for both the control and the projections. We believe that it is straightforward for the reader to substract the control if needed.

- the ISMIP6-Greenland paper shows absolute values (Goelzer et al, 2020). For consistency between our papers on Greenland and Antarctica, it makes sense to stick to absolute changes.

- the paper has been reviewed by 2 other reviewers (+2 for the Greenland paper) with no comment on this matter. Changing all the figures at this stage would ideally require the agreement of the other reviewers.

For these reasons, unless you have strong argument against it, we prefer to stick to our version of the plots showing the absolute values.

**P9 L26-27 From Figure 5 I read that only in HadGEM2-ES does surface melt, runoff and evap overcome the increase in precip? Worth stating that explicitly.**

We now make a better distinction between surface mass balance and precipitation:

"In fact, most GCMs simulate an increase in precipitation in Antarctica related to the projected warming. This increase in precipitation can be partly compensated by an increase in runoff and evaporation. However, overall, most GCMs produce an increase integrated surface mass balance in the future. The difference in terms of surface mass balance change amongst the GCMs explains the large spread in [...]".

P10 L1-2 is it really a lack in precipitation increase or rather a considerable increase in coastal surface melt and runoff? Or did you mean to write "lack of surface mass balance" here?

HadGEM2-ES produces effectively increased precipitation but not an increase in surface mass balance. We replace "precipitation" here by "surface mass balance".

P12 L28-29 but should also be associated with increased surface melt if temperatures are hight enough? How does surface mass balance change look like?

Agree, this was an oversimplification from our part. We now refer here to Nowicki et al. (2020) instead, since the changes in surface temperature and surface mass balance are presented in their paper.

P13 L6 Seroussi et al also show that ice area loss due to ice shelf collapse is 6 times larger than in the simulations without collapse (section 4.8 in their paper). This probably means that GRISLI models rather stable grounding lines compared to other models which took part in ISMIP6. This should be discussed more explicitly in the paper as it is an important piece of information for the interpretation of the results shown here.

In GRISLI, the ice shelf area when using CCSM4 under RCP8.5 is reduced by 86 000 km2 between 2015 and 2100. When using the ice shelf collapse scenario, the shelf area is reduced by 240 000 km2. The floating ice area loss due to ice shelf collapse is 2.8 larger than in the simulations without collapse. This number is smaller than in Seroussi et al. (2020), who reported a 6 times decrease in shelf area due to collapse. However, Seroussi et al. (2020) also reported a mean ice shelf area reduction of 11 000 km2 without the ice shelf collapse when using CCSM4 under RCP8.5 (compared to 86 000 km2 in GRISLI) and 66 000 km2 with shelf collapse (compared to 240 000 km2). This means that GRISLI shows a large ice shelf shrinking when compared to the ensemble spread. This higher ice shelf extent sensitivity to the oceanic forcing in GRISLI compared to the other models could explain why the ice shelf collapse scenario has a relatively lower importance in GRISLI.

We modified the manuscript in the results section. First:

"Overall, the ice shelf collapse scenario systematically induces a decrease in the ice shelf extent. For example, when using CCSM4 under the RCP8.5 the ice shelf extent decreases by 86 000 km2 from 2015 to 2100, but it decreases by 240 000 km2 with the ice shelf collapse scenario (extent loss 2.8 times larger).

A greater sensitivity to this process has been reported in Seroussi et al. (2020), although associated with a wide spread of responses amongst participating models. In terms of ice shelf extent loss, Seroussi et al. (2020) reported a loss 6 times larger with the ice shelf collapse scenario (66 000 km2 compared to 11 000 km2). However, the numbers in Seroussi et al. (2020) are much smaller than the one in GRISLI (240 000 and 86 000 km2 with and without the shelf collapse scenario, respectively) suggesting a high sensitivity of the ice shelf extent in GRISLI to the oceanic perturbation. This might explain why the ice shelf collapse has a relatively lower impact on the ice shelf extent. However, Seroussi et al. (2020) also reported a larger impact of the ice shelf collapse scenario on the volume change contributing to sea level rise (multi-model average of 28 mm SLE in 2100 under the CCSM4 forcing). This can indicate a low sensitivity of the grounding line retreat in GRISLI compared to the other participating models. However, it can also be linked to the local model biases. In fact, for most climate models, the retreat masks by 2100 [...]"

Second, as mentioned earlier in this document, a dedicated discussion to the sensitivity of the grounding line in GRISLI with respect to the other models participating in ISMIP6 has been added in the discussion section.

Also, there was a typo in the manuscript, the average impact of the shelf collapse in Seroussi et al. (2020) is 28 mm SLE (not cm SLE).

**P13 L11-13 However relatively little grounding line retreat compared to the other models which took part in ABUMIP. Actually GRISLI has the lowest volume equivalent SL change of the whole ensemble (together with PISM-PIK).**

We agree with the reviewer with the fact that GRISLI shows a relatively modest grounding line retreat when compared to other ABUMIP participating models. However, it has not the weakest ice sheet response to the loss of buttressing.

The reviewer has probably in mind Fig. 5 of Sun et al. (2020) which shows absolute volume change in ABUK (no shelf) and ABUM (artificially very high shelf melting). This figure does not account for the fact that some models show a large volume drift in their control experiments. This drift is an other manifestation of the importance of the initialisation procedure which produces an ice sheet in equilibrium with the forcings (as in GRISLI) or not. For example, PISM1 produces a volume change in ABUK of about 3 mSLE in 500 years but it also shows a large drift in the control simulation (ABUC) that leads to about 1.5 m SLE (Fig. 1). When corrected for the drift in the control, buttressing is thus accounting for about 1.5 m SLE for this model while it accounts for about 2 m SLE in GRISLI. The role of buttressing is even weaker for ISSM (which only participated to ABUM) since it produces a 1 m SLE in ABUM with a drift in the control of about 0.5 m SLE. PISM-PIK in turn has a drift towards larger ice volume and as such shows a more important role of buttressing than GRISLI (accounting for about 3 m SLE compared to about 2 m SLE in GRISLI).

To conclude, GRISLI shows indeed a relatively modest grounding line retreat induced by buttressing loss when compared to other ABUMIP participating models (3rd lowest out of 15). We added this precision in the manuscript:

" [...] we were able to simulate large grounding line retreat when the buttressing induced by the ice shelves is removed, although amongst the lowest within the other participating models (3rd lowest ice volume change with respect to the control experiment in 500 years, out of 15 participating models)"

We also have added a discussion about the GRISLI grounding line sensitivity (see our previous comment).

**P14 L24-25 why the year 2045? please explain.**

This choice is somehow arbitrary. Our idea was to test the effect of a potential change in ice mechanical parameters in the course of the century for the projections in 2100. Thus it should start not too early after the start of the simulations (for which the mechanical parameters have been tuned) nor too late (otherwise the perturbation will have no effect).

**We added the following:**

"These perturbations are imposed abruptly at the end of the year 2045, in order to mimic a potential change of these parameters in the course of the century. The timing of these perturbations is somewhat arbitrary: not too close from the start of the projections but also not too late so that they affect the ice sheet evolution to 2100."

**P14 L27 & L34 what is meant by "standard value" here?**

Unperturbed experiments. This is now specified:

"Fig. 13 shows the mass change in 2100 for the perturbed experiments with respect to their unperturbed counterpart (shown in Sec. 3.2.1)."

And later:

"For the basal drag coefficient, the acceptable perturbations lead to an additional sea level contribution ranging from about -30 to +30 mm SLE, with respect to the unperturbed NorESM1-M under RCP8.5 experiment that produces a 20 mm SLE in 2100."

And finally:

"The perturbations induce a change in total mass of  $-12 \times 10^3$  to  $+12 \times 10^3$ Gt for the basal drag coefficient and of  $-30 \times 10^3$  to  $+25 \times 10^3$ Gt for the enhancement factor, with respect to the mass loss in 2100 of  $-165 \times 10^3$ Gt obtained with the unperturbed NorESM1-M under RCP8.5 experiment".

**P15 L4 "They"**

Replaced by "The perturbations".

**P15 L30 This also means that the following two sentences don't really make sense.**

We are sorry but we do not understand your point here. What we meant is that a simple linear extrapolation of the observed recent mass loss rate gives 52 mm SLE for the Antarctic ice sheet in 2100. We do not expect a linear trend for this contribution since it exists multiple feedbacks that could lead to increased future loss rates. Since GRISLI and the majority of ISMIP6 models produce a 2100 contribution to global sea level rise lower than 52 mm SLE, there is an apparent disagreement the observed recent mass loss rate and the ISMIP6 projected contributions in 2100. We think that a large part of this disagreement comes from the initialisation procedures used by the ice sheet models since most of these procedures produce an ice sheet at equilibrium for the present-day.

P16 L9 I think at this point you should talk about surface mass balance not about precip, as SMB is ultimately the decisive variable for ice sheet mass changes.

Right, corrected.

P29 Fig. 7 caption I suggest to remove this line from this and the other figures as it is misleading. Mass changes in the AIS contribute differently to SL changes depending where the are occuring. The specification 1 mm SLE = 372 Gt does not correspond to what is shown in the figure and therefore will confuse the reader.

Agree. We put this only so that the reader can have an easy way to convert the two units but it is true that we do not used this factor to draw our plots. We removed it in the revised manuscript.

---

## Author Response (AR3)

Dear Dr. Ayako Abe-Ouchi,

thank you again for your time as Editor on our paper. We performed a careful reading to correct the remaining typographical errors. Please find in this response a track-change version of the manuscript.

Best regards,
Aurélien Quiquet and Christophe Dumas

[revised manuscript text omitted]